# Spin Qubits Candidate in Transition-Metal-Ion doped Halide Double Perovskites

Sakarn Khamkaeo[1], Kunpot Mopoung [1], Kingshuk Mukhuti[2], Maarten W. de Dreu[2], Anna Dávid [3], Muyi Zhang [1], Mats Fahlman [3], Feng Gao [1], Peter C. M. Christianen [2], Irina A. Buyanova [1], Weimin M. Chen [1] ✉ & Yuttapoom Puttisong [1] ✉

Solid-state spin qubits offer a promising route toward scalable quantum technologies. Here we demonstrate that, despites of a nuclear-spin-rich host of halide double perovskites (HDPs), transition-metal centers ($Cr^{3+}$ and $Fe^{3+}$ ions) are a good candidate for spin qubits exhibiting long-lived electron spin coherence with $T_2 = 29.5$ μs and 21.2 μs at 4 K, respectively. Notably, spin localization facilitates a well-defined electron-nuclear (e-N) spin rotation between the electron spin and the neighboring nuclear spins of $^{35,37}Cl$ and$^{133}Cs$. The resulting e-N spin cluster is readily beneficial for a target nuclear-spin sensing. For the $Cr^{3+}$ spin centers, the optical transitions associated with $Cr^{3+}$ spin centers is spin-selective thereby paving a way for optical addressing of spins. Our findings from these spin ensemble studies establish HDPs as a new promising platform for creating solid-state spin qubits using simple and inexpensive solution-based single crystal growth methods, broadening material applications of halide perovskites.

Solid-state spin qubits, utilizing localized defect and impurity centers in wide band-gap semiconductors, offer a promising route toward a scalable quantum network. These optically addressable, atomic-like spin centers can serve as local nodes for quantum manipulation[1–6]. Quantum communication is mediated by interfacing spin and photons, leveraging the selection rules of atomic-like transitions. They are highly attractive for their long spin coherence time and their compatibility with existing semiconductor technology, allowing for the integration of quantum devices with conventional electronics. The current state-of-the-art in solid-state spin qubits largely focuses on 'defects' systems like nitrogen-vacancy (NV) centers[7–13] in diamond and divacancy centers in silicon carbide (SiC)[3,14] and 'impurity' spin centers such as transition metal (TM) $V^{4+}$ and $Cr^{4+}$ centers in SiC[15–20]. These spin qubit candidates are known for their exceptionally long spin coherence times. This is partly attributed to the nature of their

semiconductor hosts, which inherently possess a majority of nuclear spin-free isotopes.

While a common design principle for solid-state spin qubits suggests using nuclear-spin-free hosts to minimize electron spin decoherence due to the interaction with a nuclear spin bath, there is a rising interest in a nuclear-spin-rich host[6–10,14–32]. These research endeavors have initially focused on expanding the spin qubit library into systems important in optoelectronics technology, such as the recent report of defect-based solid-state spin qubits in GaN[33], in transition metal dichalcogenides[34], and in hexagonal boron nitride[25,29,35]. Furthermore, recent research has highlighted the potential of electron-nuclear (e-N) hybrid systems[7,8,13,26–28,30,32,36] that utilize nuclear spins as quantum operating or registering nodes. This has been demonstrated using rare-earth impurity Yb centers within $YVO_4$[26], where the tetragonal $^{51}V$ acts as a spin-wave register for the Yb spin qubit.

[1]Department of Physics, Chemistry and Biology (IFM), Linköping University, Linköping, Sweden. [2]HFML-FELIX, Toernooiveld 7, 6525ED Nijmegen, the Netherlands; Institute for Molecules and Materials, Radboud University, Heyendaalseweg 135, 6525 AJ Nijmegen, the Netherlands. [3]Laboratory of Organic Electronics (LOE), Department of Science and Technology, Linköping University, 60174 Norrköping, Sweden. ✉e-mail: weimin.chen@liu.se; yuttapoom.puttisong@liu.se

In this context, halide perovskites are an emerging class of materials for optoelectronics and novel spintronics[37–45]. They are attractive due to their high performance and much low fabrication costs. The diverse chemical and structural varieties within the halide perovskite material library further allow for strategic design of their electronic properties[37,38], serving various applications such as solar absorbers[46], photodetectors[47], X-ray detectors[48], and white-lighting[45]. Given their potential in optoelectronics, integrating spin qubits into halide perovskite hosts could leverage existing optoelectronic and semiconductor technologies. The spin centers in HDP can be easily introduced and controlled through solution-based methods[40,49]. Additionally, the well-established solution-based growth methods offer flexibility in designing the spin and electronic states of the spin qubits. This is because the coordination partners and local symmetry around the spin qubits can be modified through coordination chemistry. While these techniques have been successfully implemented in optically addressable molecular spin qubits[50], they are relatively less explored for solid-state spin qubits.

In this work, we report spin coherent and magneto-optical properties of prototype transition metal (TM) $Cr^{3+}$ and $Fe^{3+}$ spin centers (concentration $< 10^{18}$ cm$^{-3}$) doped in $Cs_2(Na,Ag)InCl_6$ host. We adopt the concept of using TM impurities as solid-state spin qubits, an approach that has been readily adopted and successfully demonstrated in classic host materials such as SiC[15–20]. These open-shell TM centers possess a "pure" spin of $S \geq \frac{1}{2}$ in the ground state, which allows their spin sublevel manifolds to be accessed and utilized as the quantum bit states. Furthermore, TM impurities are typically optically active (or "bright") and exhibit sharp, atomic-like transitions. These properties are critical, as they permit the optical initialization and readout of the qubit's spin state by leveraging spin-optical selection rules. With a typical concentration of $< 10^{19}$, individual TM ions are single spin centers that are electronically and magnetically isolated due to their strong localization of electron wavefunctions and spin densities. Such isolation would in principle permit to address single spins by employing a sufficiently local probe, though significant

technical challenges will arise at a high concentration of spin centers when a nano-scale probe becomes necessary.

## Results

Here in, we shall show that, despite of the nuclear-spin-rich host, the studied TM spin ensemble exhibits a rather long spin coherence time $T_2 = 29.5\,\mu s$ at 4 K (6.4 μs at 20 K). This is comparable to the other benchmark solid-state spin qubits embedded in the nuclear-spin-rich host, where creating the spin qubits requires a more complicated fabrication technique[25,26]. We also observe long-lived phase undulation due to strong coupling between the electron spin at the TM centers with the first and second nearest neighbor nuclear spins. We attribute this finding to a localization of TM spin wave function, which enables a well-defined interaction with the local nuclear spin environment that could potentially facilitate nuclear spin registration protocols. For the $Cr^{3+}$ spin centers, we identify the ground and excited electronic states via magneto-optical experiments and identify their optical-transition selection rule necessary for optical addressing of the TM-spin qubit in HDPs hosts. Though limited to ensemble studies, our results shed light on a promising route towards a new library of solid-state spin qubits, thanks to the large chemical and structural diversity of halide perovskites, offering new opportunities for applications in quantum information technology potentially at a much lower cost than their alternatives.

Here we choose chromium (Cr) and iron (Fe) as TM ions in the $Cs_2(Na:Ag)InCl_6$ as our prototype spin qubits. First, unlike many halide perovskites that are prone to rapid degradation, $Cs_2(Na:Ag)InCl_6$ HDPs are wide bandgap semiconductors with exceptional stability[45]. Second, the Cr and Fe ions are thermodynamically stable in the 3+ charge state, substituting $(InCl_6)^{-3}$ by $(CrCl_6)^{-3}$ or $(FeCl_6)^{-3}$ within the cubic crystal, see Fig. 1A, B. The TM spins in this context have $O_h$ point group symmetry with an orbitally non-degenerated $A_{1g}$ ground state. This configuration significantly quenches effects of the spin-orbit interaction to the first order, beneficial for maintaining a long spin-lattice relaxation time ($T_1$) and spin coherence time ($T_2$). Additionally, the HDP crystal

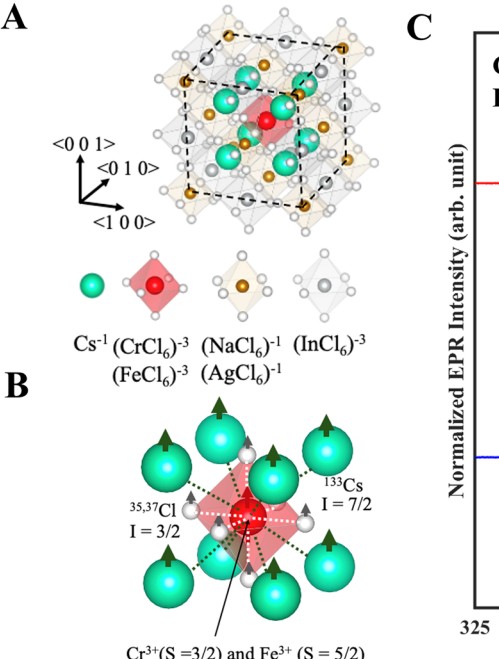

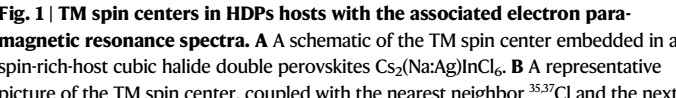

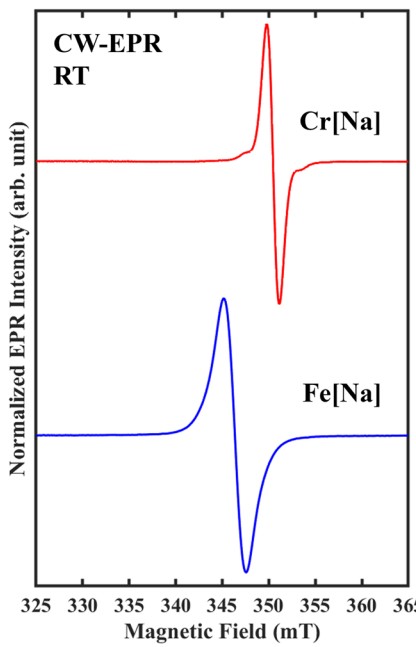

**Fig. 1 | TM spin centers in HDPs hosts with the associated electron paramagnetic resonance spectra. A** A schematic of the TM spin center embedded in a spin-rich-host cubic halide double perovskites $Cs_2(Na:Ag)InCl_6$. **B** A representative picture of the TM spin center, coupled with the nearest neighbor $^{35,37}Cl$ and the next nearest neighbor $^{133}Cs$ nuclear spins through hyperfine interactions. The white and black dotted lines represent the principal directions of the hyperfine interaction with the $^{35,37}Cl$ and $^{133}Cs$ nuclei, respectively. **C** cw-EPR spectra of Cr[Na] and Fe[Na] powder samples, measured at room temperature and a microwave frequency of 9.77 GHz.

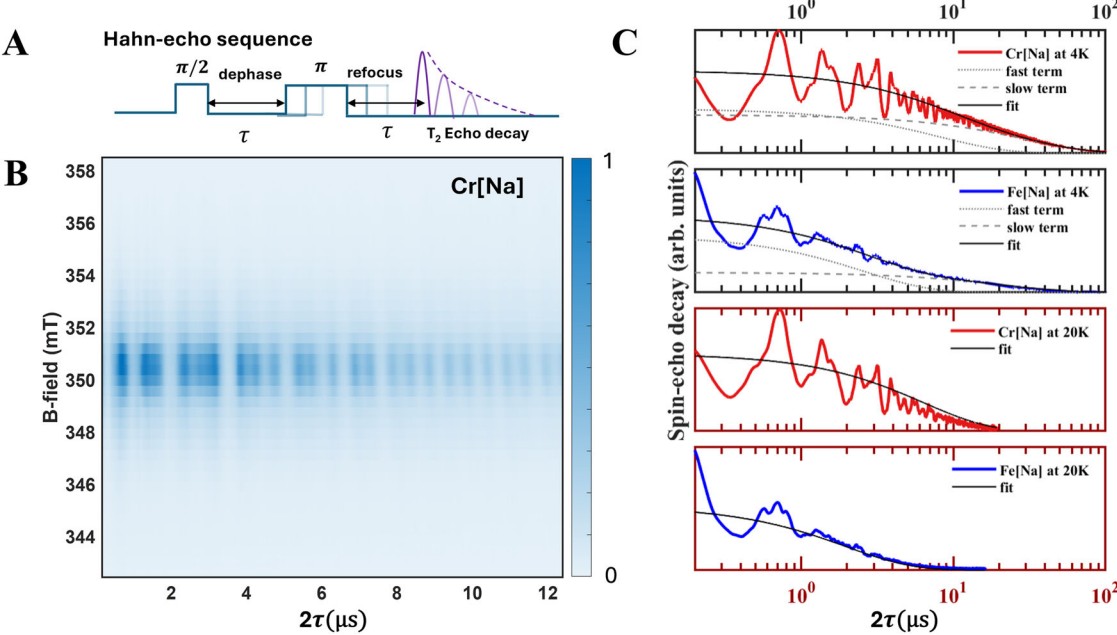

**Fig. 2 | The Hahn-echo decay of $Cr^{3+}$ and $Fe^{3+}$ spin centers in HDPs host. A** An illustration of the Hahn-echo electron spin resonance experiment, with a pulse sequence of $\pi/2 \rightarrow \tau \rightarrow \pi \rightarrow \tau \rightarrow echo$. **B** A magnetic field-sweep Hahn-echo experiment of the $Cr^{3+}$ centers in the $Cs_2NaInCl_6$ (Cr[Na]) samples performed at 4 K. The echo signal decays with time evolution $2\tau$. The central resonant field is 350.2 mT, corresponding to the $Cr^{3+}$ ions with $g = 1.981$. **C** A comparison of echo decay and

spin decoherence time ($T_2$) obtained for two different TM centers ($Cr^{3+}$ and $Fe^{3+}$) in $Cs_2NaInCl_6$ host at 4 K and 20 K. Echo decay curves were fitted with a two-exponential model at 4 K and a single-exponential model at 20 K. The individual components of the 4 K fit are shown by the gray-dotted lines for the fast decay term and the gray-dashed lines for the slow decay term. The overall fitting curves for both temperatures are represented by the solid black lines.

has the non-magnetic $B^{I}$ (+1) ions, Na and Ag, that work as a spacer, reducing self-aggregation of magnetic ions that are harmful for the spin qubits application. The (Na:Ag) hosts are chosen for monitoring the effect of the third nearest neighbor nuclear spins to the TM spin centers, which also provides a probe of the degree of wave function localization of the TM ions.

In the $O_h$ point group symmetry, the absence of a zero-field splitting results in a single electron-paramagnetic-resonance (EPR) line observed for both $Cr^{3+}$ ($S = 3/2, g = 1.981$) and $Fe^{3+}$ ($S = 5/2, g = 2.01$) TM spin centers, as shown in Fig. 1C. The nuclear spin interactions are governed by the nearest-neighbor chlorine-35,37 ($^{35,37}Cl$, I = 3/2) and the next-nearest-neighbor cesium-133 ($^{133}Cs$, I = 7/2) with six and eight lattice coordination, respectively, see Fig. 1B. The hyperfine interaction strength with these coupled nuclear spins typically falls in the range of 0.1-10 MHz. While this interaction cannot be spectrally resolved, it can significantly impact electron spin dynamics monitored in transient electron spin-echo experiments[51,52].

**Electron spin echo and electron-nuclear spin rotation**

We will now demonstrate that, despite that the $Cr^{3+}$ and $Fe^{3+}$ ions are embedded in a nuclear-spin-rich environment, a long $T_2$ can be achieved thanks to a combined effect of electron spin localization and the deterministic nature of the relevant hyperfine interaction.

To measure $T_2$, standard Hahn-echo experiments (Fig. 2A) were conducted on four representative samples: Cr and Fe ions embedded in $Cs_2NaInCl_6$ and $Cs_2AgInCl_6$ hosts, which we simply denote below as Cr[Na], Cr[Ag], Fe[Na], and Fe[Ag], respectively. All samples were powdered, and the experiments were performed using an EPR spectrometer to address ensembles of the TM spin centers. Additional details of the samples can be found in the Material and Methods section.

A representative B-field sweep echo decay of the Cr[Na] samples, showcasing the spin decoherence properties of the $Cr^{3+}$ ions, is plotted in Fig. 2B. Two striking features are observed, namely, a long

$T_2$ and a strong undulation of electron spin-echo signals. The electron spin-echo signal exhibits a non-monotonic exponential decay with fast and slow exponential components, characterized by $T_{2fast} = 8.1 \pm 0.6 \, \mu s$ and $T_{2slow} = 29.5 \pm 1.3 \, \mu s$, as fitted and shown in Fig. S1A–D. The presence of two decay components is typically observed in Hahn-echo experiments, with the fast component being associated with spin diffusion due to a possible dipolar interaction between spin centers whereas the slow component reflects the intrinsic $T_2$ for an isolated TM center. The Hahn-echo $T_2$ exceeding 20 μs, achieved even without implementing dynamic decoupling protocols, is comparable to that reported for spin centers in other potential candidates in solids with nuclear-spin-rich environment (see Table 1), where the spin qubit fabrication requires the more complicated fabrication techniques.

Figure 2C compares the coherent properties of $Cr^{3+}$ and $Fe^{3+}$, their timescale is rather similar (29.5 μs and 21.2 μs, respectively). Given that they are both orbitally non-degenerate states, we attribute the long spin coherence time to the strong wave function localization effects that limit the hyperfine interaction to the nuclear spins within a few nearest neighbor nuclei. This assignment is supported by a comparison of the spin coherent time of the same TM ions in two different hosts: $Cs_2NaInCl_6$ and $Cs_2AgInCl_6$ shown in the Fig. S2. By ensuring similar concentrations of the same TM ions in the two different host materials, we observe similar electron spin decoherence rates. Given that the third (3rd) nearest neighbor nuclear spins in the lattice position differ between the two hosts, Na, I = 3/2 vs Ag, I = ½ with markedly different nuclear magnetic moments, a similar coherent time implies that the dominant hyperfine interactions between the electron spin of the $Cr^{3+}$ or $Fe^{3+}$ ion and the surrounding nuclear spins must be limited within the second-nearest neighbor nuclei.

Upon heating up to 20 K, we obtain $T_2$ of $6.39 \pm 0.22 \, \mu s$ for $Cr^{3+}$ and $1.97 \pm 0.04 \, \mu s$ for $Fe^{3+}$ (see Fig. 2C and Fig. S1C, D). The decoherent rate displays a constant value over the temperature range of 3-10 K, which confirms that the intrinsic spin coherence time is governed by

**Table 1 | State-of-art of spin qubits**

| Material | Spin Qubit | Nuclear spin targeting | Benchmark $T_2$ | Ref. |
|---|---|---|---|---|
| **Solid state qubits** | | | | |
| Defect and impurity spin qubits | | | | |
| Nuclear-spin-rich host | | | | |
| $Cs_2NaInCl_6$ (halide double perovskites) | Cr(3 + ) and Fe(3 + ) ions | Deterministic ($^{35,37}$Cl, $^{133}$Cs) | 29.5 μs (4 K) 21 μs (4 K) | our work |
| yttrium orthovanadate (YVO$_4$) | Yb(3 + ) Ion | Deterministic ($^{51}$V) | 58 μs (500 mK) | 26 |
| Galium Oxide (Ga$_2$O$_3$) | Ti (4 + ) + e | Deterministic($^{69,71}$Ga) | 1.1 μs (4 K) | 66 |
| hexagonal Boron Nitride (h-BN) | Boron vacancy | Deterministic ($^{14}$N) | 100 ns (RT) | 25,29 |
| Galium Nitride (GaN) | Point defects | No report | 100 ns ($T_2$* RT) | 33 |
| Nuclear-spin-free hos | | | | |
| Ba2CaWO6–δ (oxide double perovskites) | Spin ½ defects | No report | 4 μs (5 K) | 67 |
| Zinc Oxide (ZnO) | Shallow Donor | No report | ~50 μs (5.5 K) | 68 |
| Diamond | NV- Center | Disorder ($^{13}$C) | Milliseconds to Seconds (RT) | 69 |
| Silicon Carbide (SiC) | Divacancy Center ($V_{Si}V_C$) | Disorder ($^{29}$Si) | ~2.3 ms (5 K) | 14 |
| Silicon (Purified $^{28}$Si) | P-Donor | Deterministic ($^{31}$P) | Up to 60 ms (300 mK) | 70 |
| Quantum dots | | | | |
| Gallium Arsenide (GaAs) | Electron | QD nuclear spin Ensemble | ~600 ns ($T_2$* 4.2 K) | 71 |
| Silicon | Electron | QD nuclear spin Ensemble | 12 μs (200 mK) | 55 |
| Germanium (Ge) | Hole | No report | 523 ns (10 mK) | 72 |
| **Molecular qubits** | | | | |
| Nuclear-spin-rich molecules | | | | |
| Cr(IV)R4 [R = o-tolyl,2,3-dimethylphenyl, 2,4-dimethylphenyl] | Cr(4 + ) | No report | 640 ns (4-5 K) | 50 |
| Cr$_7$Ni Ring | $S = 1/2$ | No report | 379 ns (4.5 K) | 73 |
| Gd@Y(trensal) | Gd(3 + ) Ion | No report | 12 μs (3 K) | 74 |
| Nuclear-spin-free molecules | | | | |
| N@C60 | N center + e | No report | 1 μs ($T_2$* RT) | 75 |
| [V(C$_8$S$_8$)$_3$]$^{2-}$ | V(4 + ) | No report | 675 μs (10 K) | 76 |

the hyperfine interaction. Above 10-50 K the rate starts to increase exponentially. This can be modelled with a localized spin-phonon interaction[53,54].

The observed strong undulation in spin echo amplitudes arises from a coherent rotation induced by the electron-nuclear (e-N) spin interaction. Here a long phase memory suggests that the phase rotation is rather homogenous, indicating that the e-N interaction is deterministic, i.e. the dominant hyperfine interaction with the surrounding nuclear spins in fixed and few lattice positions. A random placement of nuclear spins would otherwise result in a distribution of hyperfine interactions, leading to significant variations in the nuclear rotation frequency and, consequently, rapid dephasing of nuclear spin coherent rotation.

**Electron-nuclear spin coherent rotation**

The long phase memory of e-N spin coherence rotation is a consistent feature across all our samples. To evaluate the deterministic nature of these e-N spin clusters and identify the specific nuclear species driving this rotation, we conducted a nuclear frequency analysis using the spin-echo signal. We employed Hyperfine Spin-Sublevels Correlation (HYSCORE), a two-dimensional experiment, to provide a clear signature of the nuclear chemical fingerprints governing the e-N spin coherent rotation. In the HYSCORE experiment, the electron spin undergoes controlled rotations on the Bloch sphere via the specific four-pulse sequence shown in Fig. 3. During the sequence, the electron spin coherence evolves under the nuclear spin transitions of all coupled nuclei in the system. After a defined evolution period, the spin coherence is transferred between hyperfine-coupled nuclear spin states of the same nuclear species (e.g., for $S = 3/2$ system, transitions occur between $|m_s, m_I^{Cs}>$ states with $\Delta m_s = \pm 1$, but not between Cs and Cl nuclear manifolds). The resulting time-domain signal produces

a 2D interference pattern, which, after Fourier transformation, yields the nuclear frequencies in both electron spin manifolds. Therefore, what appears in the 2D contrast in Fourier transform of the HYSCORE data must be corelated within the same nuclear spin species, efficiently eliminate the complication in assigning the nuclear frequency peaks of ESEEM spectra, where the frequency spectra have multi-overlapping components. From this spectrum, the hyperfine and quadrupole tensors can be extracted, allowing precise assignment of nuclear sites. Furthermore, where the samples were in powder form as in our case, the HYSCORE pattern's sensitivity to the principal axes of hyperfine and quadrupole interactions allowed us to pinpoint the location of nuclear spins within the lattice, thereby confirming the deterministic nature of the e-N interaction.

The HYSCORE experiment exploits the pulse sequence shown in Fig. 3A. The HYSCORE spectrum of Cr[Na] powder samples (Fig. 3B) yields a two-dimensional e-N coherence spectrograph obtained from the e-N spin rotation correlated between two adjacent Zeeman sublevels. Its frequency domain spectrum in Fig. 3C, therefore, exclusively correlates nuclear frequencies within the same nuclear origin, effectively resolving the complex nuclear frequency spectrum. To attribute each nuclear contribution, we simulated the HYSCORE spectrum taking into account both electron-nuclear hyperfine and nuclear spin quadrupole interactions as shown in Fig. 3D. The simulation results confirm the contribution of both $^{133}$Cs and $^{35,37}$Cl in the e-N coherence rotation in Fig. 3B, which we differentiate using different color codes - red for $^{133}$Cs and blue for $^{35,37}$Cl.

Our simulation analysis aligns well with the underlying physics of hyperfine interactions. For the next-nearest-neighbor $^{133}$Cs nuclear spin, there's a weak interaction from the Fermi-contact term. The nuclear spin frequencies center around $\nu(^{133}Cs) \approx 1.958$ MHz and split into at least four frequencies due to the interaction between the

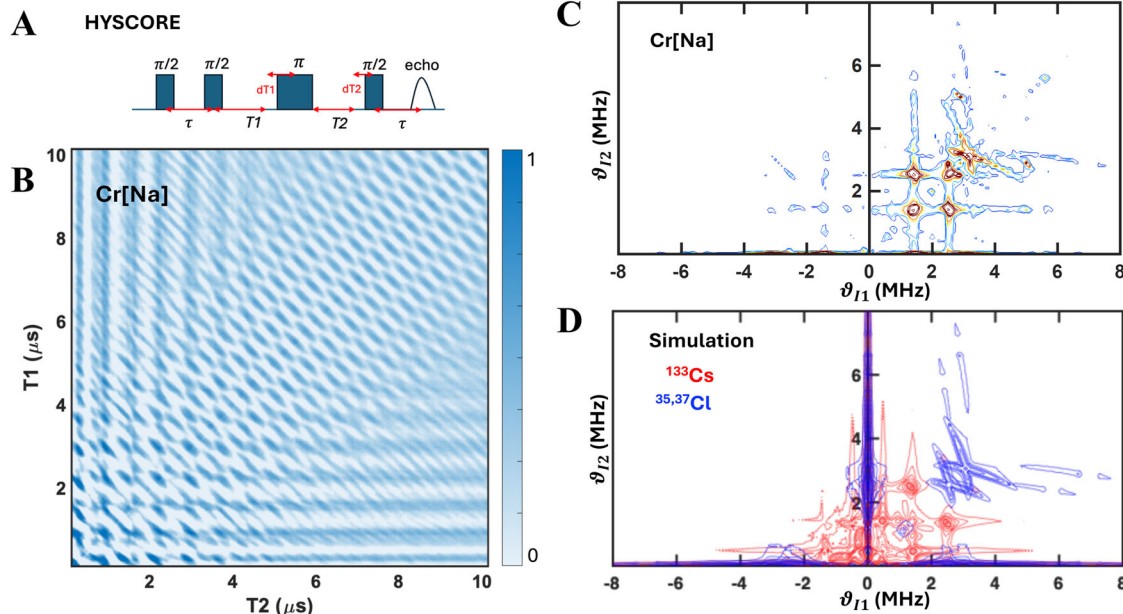

**Fig. 3 | Electron-nuclear spin coherence rotation demonstrated by HYSCORE technique in for $Cr^{3+}$ centers in $Cs_2NaInCl_6$ host. A** Pulse sequence for a HYSCORE experiment: $\pi/2 \rightarrow \tau \rightarrow \pi/2 \rightarrow T1 \rightarrow \pi \rightarrow T2 \rightarrow \pi/2 \rightarrow \tau \rightarrow$ echo. Here the second $\pi/2$ after a time delay $\tau$ partially converses electron spin coherence into polarization. This also encodes electron spin coherence to nuclear coherence under the hyperfine interaction. The $\pi$ pulse transfers spin polarization between two Zeeman sublevels, causing a relative change of the correlated nuclear spin rotation, the final $\pi/2$ transfers the electron spin polarization back to electron spin coherence, causing the stimulated echo to happen at the time $\tau$. **B** HYSCORE coherence spectrograph of the Cr[Na] samples at 4 K, mapped in two dimensions as a function of T1 and T2. **C** Frequency domain plot of the HYSCORE data. **D** Nuclear frequency analysis, including the contribution of $S = 3/2$ $Cr^{3+}$ interacting with both $^{133}Cs$ ($I = 7/2$) and $^{35,37}Cl$ ($I = 3/2$) nuclear spins.

---

electron spin quantum numbers ($m_s = \pm1/2, \pm3/2$) and the hyperfine tensor $\mathbf{A} = (0.94, 0.94, 1.42) \pm 0.3$ MHz, with the principal axis along < 111 >, identifying that the $^{133}Cs$ is indeed at their $2^{nd}$ nearest neighbor lattice site. No further splitting of nuclear frequencies is observed because $^{133}Cs$ has an intrinsically weak quadrupole moment. The HYSCORE powder simulation for these given parameters is shown in red in Fig. 3D. $^{35,37}Cl$ nuclear frequencies are influenced by both the hyperfine interaction with $Cr^{3+}$ ions and the quadrupole interaction within the $I = 3/2$ nuclear spin. The best fit, using the spin Hamiltonian (blue color in Fig. 3D), yields $\mathbf{A}(^{35}Cl) = (0.42, 0.42, -7.12) \pm 0.3$ MHz, the quadrupole tensor $\mathbf{Q}(^{35}Cl) = (-2.01, -2.01, 4.02) \pm 0.3$ MHz, $\mathbf{A}(^{37}Cl) = (0.35\ 0.35\ -5.93) \pm 0.3$ MHz, and $\mathbf{Q}(^{37}Cl) = (-1.68, -1.57, 3.35) \pm 0.3$ MHz, with principal axis along <001> for both A- and Q-tensors, i.e. the octahedral lattice site of $^{35,37}Cl$. Our determined spin parameters of $Cr^{3+}$ in $Cs_2NaInCl_6$ are also in line with previous reports from CW-ENDOR experiments[55]. The hyperfine parameters reflect the d-orbital wave function of the TM ions, while presence of the quadrupole coupling is expected due to the gradient of the electron wave function across the Cl site, given the Cr-Cl-Na imbalance in electronegativity. Our powder fitting reproduces the dominant feature, which is a crossing point around $v_1 = v_2 = 3.2$ MHz, with a large spread of matching nuclear spin rotation frequencies due to the anisotropy of $\mathbf{A}$ and the presence of $\mathbf{Q}$. We stress that the long e-N coherent rotation, as seen in the coherent ripple of HYSCORE pattern, is important in designing EPR pulsed sequence of e-N spin transfer for nuclear spin register protocol[28]. Such information is lacking in the previous CW-ENDOR study[56], where the resonant nuclear spin transition is a semi-classic spin-flip process.

HYSCORE experiments and fitting evaluations were performed on all samples (Fig. S3A–F and Table S1). The results reveal a generally weaker hyperfine interaction between the $Fe^{3+}$ electron spin and the $^{133}Cs$ nuclear spin, causing all Cs-related nuclear spin frequencies to collapse to the fundamental Larmor frequency of $^{133}Cs$ (1.9458 MHz). Additionally, the quadrupolar coupling of both $Fe^{3+}$ and $Cr^{3+}$ at the $^{35,37}Cl$ sites is generally reduced in the $Cs_2AgInCl_6$ host as compared with the $Cs_2NaInCl_6$ host, as expected from a smaller electric field gradient across the Cr-Cl-Ag and Fe-Cl-Ag bonds in the former.

With the given hyperfine parameters, we can evaluate the degree of electron localization using the Fermi hyperfine contact model, shown in the Supporting Information on Fig. S4. We found that the electron wavefunction is effectively diminishing below 0.02% at the $^{133}Cs$ sites.

## Target coupling of electron-nuclear spins in single crystalline samples

To address a target nuclear spin deterministically, we choose $^{133}Cs$ instead of $^{35,37}Cl$ nuclear spin, as a random presence of two different isotopes of Cl on the nearest-neighbor sites will lead to a non-deterministic variation between the TM ions. As a spin registering protocol is based on coherent rotation between e-N spin governed by a non-secular term in spin Hamiltonian. In single crystalline samples, this coherent mixing will happen only when the principal axis of electron/nuclear Zeeman terms are off the principal axis of the hyperfine and quadrupole interaction. We provide additional explanation of these in Supporting Information.

Therefore, we can minimize the spin mixing contribution of e-N spin interaction with $^{35,37}Cl$ by applying magnetic field along the [001] direction in the single crystalline HDP samples, as shown in Fig. 2A, which are the principal axis of A- and Q-tensor of $^{35,37}Cl$. This way the echo-undulation by the Cl-nuclear spin is minimized, leaving the e-N spin rotation exclusive to the Cs suitable for target nuclear spin registration.

In Fig. S5A–D, we present three-pulsed stimulated echo spectra of Cr[Na] in high-quality single crystals from the one-dimension HYSCORE experiment and a nuclear frequency analysis with the external B-field aligned close to the [001], [111], and [011] crystallographic axes. The observed nuclear frequencies at each orientation are well described by the spin-Hamiltonian parameters of $^{133}Cs$ and

$^{35,37}$Cl obtained in the HYSCORE experiments. Notably, at B// < 001 >, the contribution from $^{35,37}$Cl in e-N spin mixing is suppressed as expected. Therefore, only the $^{133}$Cs nuclei become the dominant source to e-N coherent rotations.

The nuclear frequency spectra with the target $^{133}$Cs nuclear spin in the Cr[Na] sample are plotted in Fig. 4. Here we identify six dominant nuclear frequencies of $^{133}$Cs coupled with the electron spin $S = 3/2$, namely $\vartheta_x^{m_s}$, where $m_s = \pm\frac{1}{2}, \pm\frac{3}{2}$ is the electron spin quantum number and x denote the type of electron-nuclear spin rotation, which involves: (1) *I*: a fundamental nuclear Larmor frequency of $^{133}$Cs (1.9458 MHz); (2) *SQ*: the allowed single quantum transition ($\Delta m_s = \pm 1, \Delta m_I = \pm 1$); and (3) *NDQ*: the forbidden nuclear double quantum transition respectively ($\Delta m_s = \pm 1, \Delta m_I = \pm 2$). We further discuss this nuclear rotation frequencies based on the energy diagram shown in Fig. S6.

Notably, each frequency further splits into four peaks, as shown in the insets. This splitting arises from the four magnetically inequivalent sites among eight $^{133}$Cs sites in the second nearest neighbor shell of the Cr$^{3+}$ ion when the magnetic field (B) is slightly misaligned from the [001] direction, making them distinguishable in the frequency spectrum (see simulations in Fig. S5A–D). The ability to identify magnetically inequivalent sites of $^{133}$Cs suggests that each Cs pairs can work as a distributable nod of target nuclear spin registration.

## Coherent control and addressing of nuclear spin through electron-nuclear spin phase rotation

The studied TM spin centers can be coherently driven in a nutation experiment, as shown in Fig. 5A, B. We plot the Rabi rotation, obtained from the powder sample Cr[Na], and Rabi oscillation frequency ($\omega_{Rabi}$) as a function of the driving microwave frequency ($\omega_1$). We here clarify that the driving power ($\omega_1 = 0 - 20$ MHz) is different from the microwave resonance frequency ($\omega_r = 9.77$ GHz). The resonance frequency is the fixed energy required for a spin flip, whereas the driving power is the oscillation rate of the spin state driven by the microwave pulse. Under hard-pulse conditions $\omega_1 > 12$ MHz, we observe the expected Rabi oscillations of the TM spin qubits at $\omega_{Rabi} = \omega_1$. However, as the driving frequency decreases, $\omega_{Rabi}$ exhibits a multi-peak behavior. This can be understood by considering the interplay between electron-electron self-interaction and electron-nuclear spin interactions, which introduce additional driving forces for Rabi rotations[57–59]. This interplay is captured by the rotating-frame spin Hamiltonian:

$$\hbar\omega_{Rabi} = (g + \Delta g)\mu_B B_1 S_x + D\left(S_z^2 + \frac{1}{3}(S(S+1))\right) + SAI + P\left(3I_z^2 + I(I+1)\right) \quad (1)$$

Here, $\Delta g$ describes the inhomogeneous broadening of the TM electron spin, accounting for the fast-dephasing decay $T_2^*$ and $\omega_{Rabi}$ broadening at large $\omega_1$. P is associated with Q tensor via Q = (-P/3, -P/3, 2P/3). When $\omega_1$ is comparable with the zero-field splitting $D$, the multi-peak $\omega_{Rabi}$ is dominated by electron-electron spin interactions. The resulting Rabi rotations are originated from $S = 3/2$[59]. The strength of the D tensor can be estimated from the vanishing of the multipeak and the emergence of the fundamental frequency $\omega_{Rabi} = \omega_1$. This yields $D$ ~ 12 ± 2 MHz, suggesting that the Cr$^{3+}$ ions are slightly displaced from the ideal cubic crystallographic site. At low driving power (1.5 MHz), the e-N spin interaction also contributes to $\omega_{Rabi}$, resulting in characteristic nuclear frequency peaks with the frequency in the range of 1- 4 MHz[60].

Finally, we demonstrate nuclear spin sensing experiments for the studied e-N spin clusters in the HDPs. This is achieved using standard nuclear sensing methods based on Carr-Purcell-Meiboom-Gill (CPMG) pulse sequences[61,62]. Previous work has shown that the modified CPMG pulse sequences, such as Pol-CPMG[28], can be used to register and read

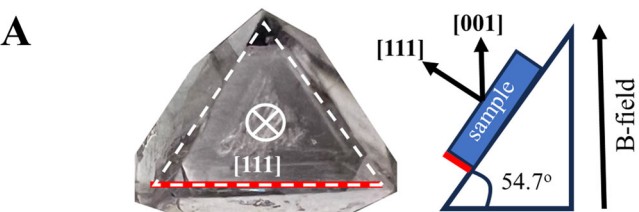

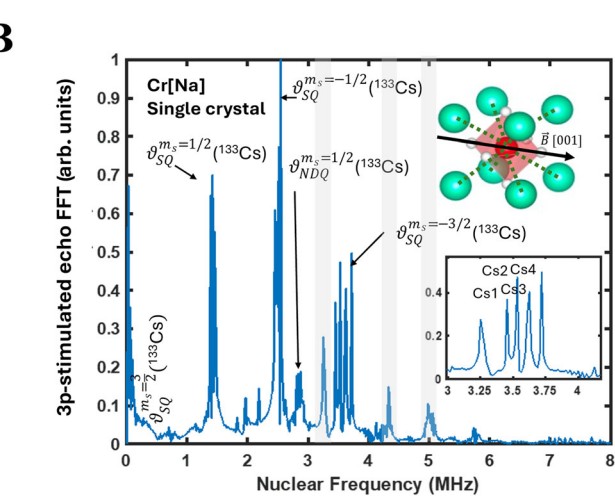

**Fig. 4 | Deterministic addressing the $^{133}$Cs nuclear spin in the single crystal experiment. A** The single crystal Cr[Na] sample and the experiment set up that orients the B-field along the [001] crystallographic direction. The triangular surface of the crystal is the (111) surface. **B** Nuclear frequency analysis based on the three-pulse stimulated echo experiments (3P-ESEEM) performed on a Cr[Na] single crystal with the magnetic field (**B**) aligned close to the [001] direction selectively coupling to the $^{133}$Cs nuclear spins. All nuclear frequencies associated with $^{133}$Cs are identified through simulation fitting. The inset highlights the splitting of the peaks corresponding to four magnetically inequivalent Cs sites. This splitting arises from a slight misalignment of the B-field from the [001] direction, which also causes a minor reappearance of $^{35,37}$Cl nuclear frequency peaks (the gray shaded area). See Fig. S5A–D, for further simulations clarifying this effect.

out nuclear spin states. This technique relies on the nuclear magnetic resonance (NMR) effect in CPMG dynamic decoupling[28,61].

The standard CPMG experiment, with the sequence $\pi_x/2 \longrightarrow^{\tau/2} [\pi_y \longrightarrow^\tau]^N \pi_y \longrightarrow^{\tau/2} echo$, where $N$ is the number of $\pi_y$ refocusing pulse, is shown is Fig. 5C. A common misconception is that the CPMG sequence is solely a "dynamic nuclear spin decoupling protocol" that refocuses random nuclear spin baths. However, we stress that this is not the case for specific nuclear spins at fixed positions near the spin centers. In our experiment in single crystals (B//[001]), the fixed $^{133}$Cs spins on the second-nearest-neighbor sites cannot be effectively decoupled. Instead, when the inter-pulse delay, $\tau$, matches a multiple of the inverse of the nuclear Larmor frequency

$$\tau = n/\vartheta_N$$

the refocusing pulses become ineffective. This condition, known as an NMR dip (as shown in the Fig. 5C–E), actually reinforces the electron-nuclear spin interaction rather than decoupling it. These resonance dips are the key for nuclear spin sensing experiments[63].

In our CPMG results (single crystal of Cr [Na], B//[001]), we exclusively observe dips from the $^{133}$Cs nuclear spins. As shown in the Fig. 5E, we demonstrate that we can sense the $^{133}$Cs nuclear spin rotation at three of the four electron spin sublevels by performing a Fourier transformation of the CPMG coherent contrast. This is shown in the results and spin structure below. Furthermore, the long $T_2$ time at 20 K

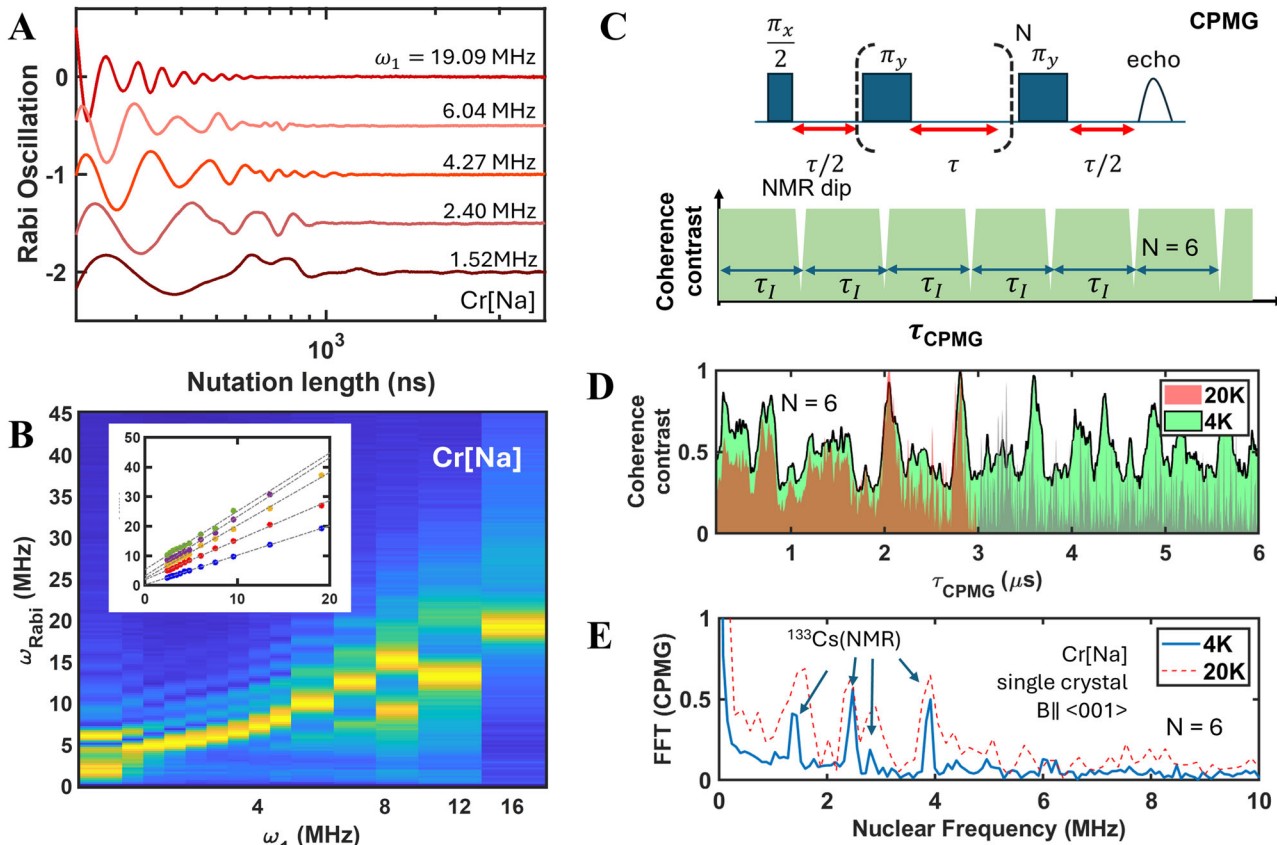

**Fig. 5 | Coherent manipulation and sensing of electron and nuclear spins in a Cr[Na] single crystal. A** Rabi oscillation of spin qubits in Cr[Na] under different microwave driving frequencies ($\omega_1$). **B** Rabi frequency ($\omega_{Rabi}$) as a function of $\omega_1$, showing complex behavior due to the interplay between microwave driving, electron spin-spin self-interaction ($S = 3/2$), and electron-nuclear hyperfine interaction. **C** A Carr-Purcell-Meiboom-Gill (CPMG) pulse sequence for nuclear spin sensing experiments. The dashed brackets represent the N repetition of $\pi_Y$ refocusing pulse. The nuclear magnetic resonance (NMR) dips are periodic with the resonant

nuclear rotation frequency with the period $\tau_I$. **D** Coherence contrast (the CPGM echo/envelop decay) as a function of the CPMG inter-pulse delay ($\tau_{CPMG}$) for the Cr[Na] single crystal with the magnetic field (**B**) aligned close to the [001] crystallographic direction, measured at 4 K (green) and 20 K (red). The gray data set is the 20 K results that are below the noise levels. **E** NMR frequency spectrum obtained by Fourier transformation of the CPMG coherence contrast shown in (**D**) demonstrating nuclear spin sensing at 4 and 20 K. The marked positions indicate NMR frequency of $^{133}Cs$ that interacts with $Cr^{3+}$ spin qubit centers.

enables CPMG nuclear spin sensing experiments at this temperature, as shown in Fig. 5D-E. This experiment thus identifies the key elements for target nuclear spin addressing within our system.

To justify that our identification of CPMG resonant dips is a beneficial first step toward target nuclear spin registers, we refer to the work of Pol-CPMG by J.E. Lang et al.[11]. The Pol- CPMG adjusts the $\pi_Y$ pulses to $\pi_Y + \theta$, one can induce polarization transfer between the electron and nuclear spin manifolds. This transfer is reflected in both the splitting and the polarization of the CPMG resonance dips, demonstrating a path toward controlled spin registers. We attempted to achieve nuclear spin registering using this polarization-transfer CPMG (Pol-CPMG) technique. However, as also noted in the Lang et al. work, this is a dynamic process that requires a large number of pulse sequence repetitions. Due to experimental limitations, our sequence was limited to $N = 6$, which was insufficient to demonstrate effective polarization transfer between the electron and nuclear spins. Nonetheless, our results represent a fundamental first step. By successfully demonstrating nuclear spin sensing—the ability to identify and detect specific nuclear spins—we have accomplished a necessary prerequisite for achieving nuclear spin registers.

## Optical addressability of the TM spins in HDPs

Here we show that the studied TM spin centers can be optically accessed – a property required for a solid-state spin qubit that is also a prerequisite for potential spin-photon interface. This is achieved via

the intra-d-shell optical transitions of the TM ions. We demonstrate this with the $Cr^{3+}$ ions in a single crystal of $Cs_2InNaCl_6$, of which the electronic structure is depicted in 6 A. Besides the $S = 3/2$ ground state $^4A_{1g}$ studied in our EPR studies, the $^4T_{2g}$ excited state splits into four sublevels due to the spin-orbit interaction. The lowest of these excited states is the $\Gamma_{7,e}$ ($m_j = \pm 1/2$) doublet. The optical transition from this lowest excited state to the ground state, i.e. $\Gamma_{7,e} \rightarrow \Gamma_{8,g}$, gives rise to a sharp zero-phonon photoluminescence (PL) line at 841.4 nm at a cryogenic temperature as displayed in the right panel of Fig. 6B. The optical transitions involving higher-lying excited states are revealed in our PL excitation experiments, see the left panel of Fig. 6B.

From the predicted selection rules, see Fig. 6A, the $\Gamma_{7,e} \rightarrow \Gamma_{8,g}$ optical transition should be highly spin selective, providing a mean for optical access to the concerned spin states that could enable their initialization and readout. This is confirmed from our polarization-resolved magneto-optical studies of the $\Gamma_{7,e}$ ($m_j = \pm 1/2$) $\rightarrow \Gamma_{8,g}$ ($m_s = \pm 1/2, \pm 3/2$) PL line, as shown in Fig. 6B. In an applied magnetic field, the 841.4 nm PL line clearly splits into several components due to the Zeeman effect. In the Faraday geometry at 2.2 K, only transitions from the lower Zeeman sublevel of the lowest excited state, $\Gamma_{7,e}$ ($m_j = -1/2$) to the two ($m_s = -3/2, + 1/2$) of the four sublevels of the ground states $\Gamma_{8,g}$ are observed with $\sigma^+$ (red) and $\sigma^-$ (blue) polarization, respectively. The $\pi$ component associated with the $\Gamma_{7,e}$ ($m_j = -1/2$) $\rightarrow \Gamma_{8,g}$ ($m_s = -1/2$) transition is only weekly observed between these two main peaks, which is expected in the Faraday geometry. Further experimental

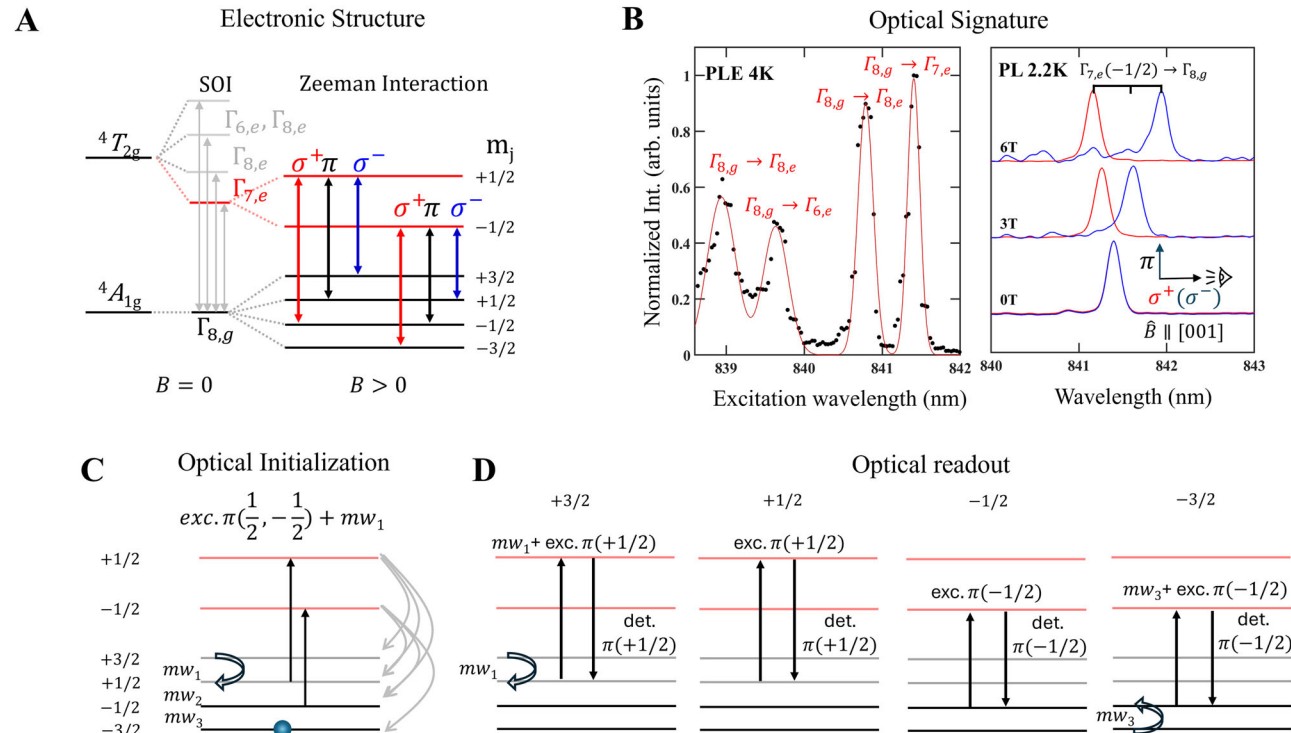

**Fig. 6 | Electronic structure of $Cr^{3+}$ centers in $Cs_2NaInCl_6$ hosts in magnetic field and optical initialization/readout protocols. A** The electronic structure and associated optical transitions of $Cr^{3+}$. **B** Spin-selective optical signature of the $Cr^{3+}$ spin centers in $Cs_2NaInCl_6$. **C** Protocol for high-fidelity optical initialization to the $|-3/2\rangle$ state and **D** for optical readout of each spin state.

proof for these spin-selective transition rules from angle- and temperature-dependent magneto-PL up to $B = 16$ Tesla is provided in the Supporting Information in Fig. S7A–E. A detailed analysis of these magneto-optical results also yields the g-values of both ground and excited states, namely, $g(\Gamma_{8,g})=1.981$ that is the same as that determined from the EPR study and $g(\Gamma_{7,e}) = 2.85$.

The successful determination of these optical selection rules allows us to propose a protocol for the optical initialization and readout of the $Cr^{3+}$ spin centers, as shown in Fig. 6C, D. The high-fidelity optical initialization into the $\Gamma_{8,g}$ ($m_S = -3/2$) state can be achieved through an optical pumping cycle shown in Fig. 6C. While the $\Gamma_{8,g}$ ($m_S = \pm1/2$) states can be directly depleted by the π-polarized resonant excitation light to the $\Gamma_{7,e}$ ($m_j = \pm1/2$) excited states, the depletion of the $\Gamma_{8,g}$ ($m_S = +3/2$) requires a combination of the resonant π-polarized laser light associated with $\Gamma_{8,g}$ ($m_S = +1/2$) and a resonant microwave field $mw_1$ that links $\Gamma_{8,g}$ ($m_S=+3/2$) with $\Gamma_{8,g}$ ($m_S = +1/2$). The subsequent recombination processes will partially convert the spin centers originately sitting in these three states to $\Gamma_{8,g}$ ($m_S=-3/2$) that cannot be removed by the optical pumping. Continuous optical pumping will eventually drive nearly all spin centers to the desired $|-3/2\rangle$ state, after a sufficient number of the pumping and recombination circles. We note that this optical initialization process is similar to that successfully applied to the $S = 3/2$ spin qubits based on the negatively charged silicon vacancy in SiC[64], but with an added advantage in our case where the $\Gamma_{8,g}$ ($m_S = +1/2$) and $\Gamma_{8,g}$ ($m_S = -1/2$) states can be separately and individually addressed as the energies of their π-polarized optical transitions are different due to the difference in the g-values between the ground and excited states, i.e. $g(\Gamma_{8,g}) = 1.981$ vs. $g(\Gamma_{7,e}) = 2.85$.

After initialization, spin manipulation is performed with microwave pulse trains using a combination of frequencies $mw_1$, $mw_2$, and $mw_3$. This allows us to fully control the entire four-state system ($m_S = \pm1/2, \pm 3/2$), utilizing the high electron spin to realize a qudit, which is a multi-level quantum bit.

Finally, state-selective optical readout can be accomplished by probing the state occupation of each spin sublevel, as shown in Fig. 6D. This is done by mapping each state's occupation onto the π-polarized emission signals. For example, to detect the population of the $|+1/2\rangle$ state, we measure its π-polarized fluorescence directly under the corresponding π-polarized resonant excitation. To detect the population of another state like $|+3/2\rangle$, we first apply a microwave π-pulse ($mw_1$) that transfers spin from the $|+3/2\rangle$ state to the $|+1/2\rangle$ state followed by a subsequent fluorescence measurement of the $|+1/2\rangle$ state as described above. This "swap-and-measure" technique can be applied to all sublevels, giving us a complete picture of the final quantum state of the spin qubit.

A key requirement for this protocol to work is that the microwave frequencies ($mw_1$, $mw_2$, $mw_3$) that manipulate spin between all adjacent spin sublevels must be distinguishable. This means that a Zero-Field Splitting (ZFS) of the $S = 3/2$ ground state is required. We show that it's possible to create this splitting through local-symmetry engineering. By alloying the $Cs_2NaInCl_6$ host with 10% $Ag^+$ on the $Na^+$ site, some of the $Cr^{3+}$ centers experience a local tetragonal distortion. This distortion breaks the cubic symmetry and lifts the degeneracy of the $m_S = \pm1/2$ and $m_S = \pm3/2$ sublevels, creating the exact ZFS we need (see Fig. S8). This result highlights the chemical tunability of the double perovskite lattice, offering a powerful way to engineer the spin Hamiltonian for advanced quantum applications. The vast chemical and structural diversity of halide double perovskites make them an ideal sandbox for designing and engineering the next generation of spin qubits and qudits.

We note that, for the $Fe^{3+}$ spin qubits centers, we unfortunately cannot so far observe the emission related to the $Fe^{3+}$ intramolecular transition. We attribute this to some external effect such as formation of competing non-radiative recombination channels due to the Fe incorporation into the HDP hosts, rather than a limit inherent to the $Fe^{3+}$ ions. Future research efforts are required to uncover the concerned optical transition by improving material quality.

The successful identification of the optical selection rules confirms the viability of this system for the initialization and readout of spin information. While scaling to the single-spin and single-photon manipulation required for quantum communication is beyond the scope of this work, our results establish a clear pathway. Achieving this goal will necessitate further advances in materials synthesis to control the concentration and positioning of TM ions, a research direction we are actively pursuing. Nonetheless, this ensemble-level demonstration of robust spin properties and a spin-optical selection rule validates this halide double perovskite material library for quantum applications and will undoubtedly drive further research in the field.

In summary, we explore the potential of halide perovskites as a platform for solid-state spin qubits. By embedding Cr and Fe ions within the double perovskite $Cs_2(Na:Ag)InCl_6$, we have demonstrated not only a long spin coherence time despite the nuclear-spin-rich host but also the deterministic nuclear spin addressing. This is enabled by the localized nature of the electron spins and the well-defined hyperfine interactions along specific crystallographic directions. We have also shown that the spin states of the TM ions in HDPs are optically addressable for their initialization and readout. Such optical access could potentially be explored for spin-photon interfacing that is attractive for creating long-range spin qubit network and for quantum communication. These findings suggest that halide perovskites, with their chemical and structural diversity, represent a highly promising platform for developing scalable spin qubits. This, combined with superior optical properties and inexpensive material synthesis of the diverse halide perovskite family, could open up many attractive and promising opportunities of applications in optoelectronics, spintronics, spin-photonics and quantum information technology.

## Methods

### Sample preparation

Single crystals of Cr- and Fe-doped $Cs_2NaInCl_6$ and $Cs_2AgInCl_6$ alloys were synthesized using a hydrothermal method. The starting materials of CsCl, AgCl, NaCl, $CrCl_3$, $FeCl_3$, and $InCl_3$ were dissolved in HCl using a stochastic formula ratio and then transferred to a Teflon-lined autoclave. The introduction of TM ions was done by replacing the $InCl_3$ precursor with the small molar ratio between $CrCl_3(FeCl_3):InCl_3$ in the range of 1–5%. A sealed autoclave was heated to 180 °C for 12 h and then cooled down to room temperature at 1 °C per hour.

The single crystals were ground to a powder form for spin-echo, Rabi rotation and HYSCORE experiments. The as-grown single crystals were used for 3P-stimulated echo, CPMG, PLE and magneto-PL experiments, where custom-made sample holders were used to align a specific crystallographic direction along the direction of an applied magnetic field.

Due to the different incorporation rates of isolated $Cr^{3+}$ and $Fe^{3+}$ ions into the two distinct host crystals ($Cs_2NaInCl_6$ and $Cs_2AgInCl_6$), electron paramagnetic resonance (EPR) spectroscopy was employed to monitor and precisely control their concentrations through spin-counting experiments. The $Cr^{3+}$ concentration in the Cr[Na] and Cr[Ag] samples was found to be ~0.3-1×10^{21} cm^{−3}, while the $Fe^{3+}$ concentration in the Fe[Na] and Fe[Ag] samples was higher, at ~0.5-1×10^{22} cm^{−3}, see Fig. S9. This observed difference is attributed to the more efficient incorporation of $Fe^{3+}$ into the $Cs_2NaInCl_6$ and $Cs_2AgInCl_6$ host lattices.

### Electron Paramagnetic Resonance Spectroscopy (EPR)

Pulsed X-band EPR measurements were performed using a Bruker ELEXSYS ESP 380 FT-EPR spectrometer equipped with a Split Ring Resonator (ER 4118X-MS5) and a closed-cycle helium gas siphon cooling (Stinger) cryostat. Large single crystals (3–12 mm lateral size) were ground into a fine powder, transferred to a 4 mm diameter quartz tube, evacuated to ~10^{-2} mbar, and flame-sealed. All measurements were performed at 4 K using the powdered samples unless otherwise stated in the main text.

Detailed descriptions of the pulsed EPR techniques used in this work, including Hahn-echo, HYSCORE, three-pulse stimulated echo, echo-detected electron spin nutation, and CPMG for nuclear spin sensing can be found in the Supporting Information, in Figs. S10–14.

### Photoluminescence Excitation (PLE) Spectroscopy

PLE measurements were performed at 4 K using a cold-finger cryostat. A wavelength-tunable Ti:Sapphire laser served as the excitation source. The resulting fluorescence was monitored at the phonon sideband of the $Cr^{3+}$ emission and analyzed with a spectrometer coupled to a silicon CCD array.

### Magneto-Photoluminescence (Magneto-PL) Spectroscopy

Magneto-PL experiments were conducted at the High Field Magnet Laboratory (HFML) in Nijmegen, the Netherlands. Samples were placed in a cryogenic bath for measurements at 2.2 K (or at 15 K using a heater). Custom holders were used to align the crystal's [001] and [111] axes parallel to the magnetic field. A 532-nm laser provided the excitation, and the emitted photoluminescence was analyzed by a spectrometer and a silicon CCD array. To measure the emission's circular polarization, a quarter-wave plate ($\lambda$/4) and a linear polarizer were placed together at the entrance of the spectrometer. The detection was in a Faraday geometry.

### Simulation

All the simulations were performed using the EasySpin[65] toolbox developed for MATLAB, setting the spin Hamiltonian as follows:

$$H = g_e\mu_B\vec{B}\cdot\hat{S} + g_N\mu_N\vec{B}\cdot\hat{I} + \hat{S}\cdot D\cdot\hat{S} + \hat{S}\cdot A\cdot\hat{I} + \hat{I}\cdot Q\cdot\hat{I}$$

All the fitting parameters used in the simulation are described in the main text, and the Table S1.

## Data availability

All the data used to produce the figure and analysis in this article are deposited at Zenodo https://doi.org/10.5281/zenodo.17882994. The data used in Supporting Information is available upon request.

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

## Acknowledgements

This work was financially supported by the Swedish Research Council (Dnr. 2021-05790), the Knut and Alice Wallenberg Foundation (Dnr. KAW 2019.0082), the Swedish Energy Agency (Dnr. 48758-1 and Dnr. 48594-1), and the Swedish Government Strategic Research Area in Materials Science on Functional Materials at Linköping University (Faculty Grant SFO-Mat-LiU No. 2009-00971). The measurements in high magnetic fields were supported by HFML-FELIX, member of the European Magnetic Field Laboratory (EMFL) and the EU Horizon 2020 EMFL-ISABEL secondment program.

## Author contributions

Y.P. and W.M.C. conceive the idea for spin qubits in halide double perovskites. S. K. and Kunpot M. performed pulsed electron-spin-resonance measurements and analysis data under the supervision of I.A.B, W.M.C. and Y.P. Kingshuk M. and M. de D. performed magneto-PL under supervision of P.C.M.C. Y.P. performed PLE. The hydrothermal growth of all samples was performed by A.D. and M.Z., under the supervision of M. F. and F. G. Y. P. wrote the paper with the helps from S.K., Kunpot M., I. A. B. and W. M. C.

## Funding

## Competing interests

All authors declare no competing interests.
