## [Transparent Peer Review file · Nature Communications]

Spin Qubits Candidate in Transition-Metal-Ion doped Halide Double Perovskites

Corresponding Author: Dr Yuttapoom Puttisong

Version 0:

Reviewer comments:

Reviewer #1

(Remarks to the Author)

In this work the authors perform ESR spectroscopy on ensembles of Chromium (Cr^{3+}) and Iron (Fe^{3+}) ions in a nuclear spin rich perovskite ($\text{Cs}_2(\text{Na:Ag})\text{InCl}_6$). They utilize several standard pulse sequences to characterize the coherence times of the Cr^{3+} and Fe^{3+} electron spins and various Hamiltonian parameters (such as the hyperfine tensor, A , and quadrupole tensor, Q). My three main criticisms of this work are the following:

1. I do not believe that the platform will be viable or competitive for the applications listed by the authors.
2. Several of the authors' Hamiltonian characterization results were already demonstrated in a previous work [Akhmin, S. M. et al., "Ligand ENDOR of Cr^{3+} and Yb^{3+} ions in $\text{Cs}_2\text{NaInCl}_6$ ", *Phys.: Condens. Matter* (1990)].
3. The paper, in multiple places, greatly exaggerates the impact of the authors' results.

For these reasons I cannot recommend publication of this work in Nature Communications. I provide more detail on each of these criticisms below.

Regarding Applications of The Platform:

The authors claim that the applications of this platform are for "scalable quantum networks and complex quantum logic gates". However, in my opinion neither of these are well justified:

1. The authors haven't demonstrated addressability of individual electron spins (i.e. qubits), only ensembles.
2. They haven't demonstrated (or even discussed) a coherent optical interface required for networking.
3. The authors haven't demonstrated the ability to initialize (or optically pump) the spins.
4. If used exclusively as a quantum processor (i.e. no coherent optical interface) this platform would be extremely limited with only a handful of qubits, therefore it could never compete with systems such as neutral atom arrays.

Regarding Novelty of This work:

There are two main results comprising this work. The first: measurement of spin echo coherence times (in the section "Electron spin echo and electron-nuclear spin entanglement") is novel. The second: characterizing Hamiltonian parameters (in the section "Deterministic hyperfine interaction"), however, is not. Several of these parameters were already measured using similar techniques in [Akhmin, S. M. et al., "Ligand ENDOR of Cr^{3+} and Yb^{3+} ions in $\text{Cs}_2\text{NaInCl}_6$ ", *Phys.: Condens. Matter* (1990)], which the authors do cite. One key distinction between these papers is that the present work considers multiple materials: both Cr^{3+} and Fe^{3+} in $\text{Cs}_2\text{NaInCl}_6$ and $\text{Cs}_2\text{AgInCl}_6$ were investigated. However, the coherence properties for these four material systems are extremely similar and the authors haven't explained why different platforms should be considered or compared their advantages/disadvantages. The final two sections of the paper mostly validate the findings of the first two sections and, in my opinion, do not contribute significantly to the novelty of the work.

Regarding Exaggerated Claims:

There are multiple places where the authors exaggerate the impact of their work

- On line 137 the authors state that "This coherence time remains exceptional even when compared to various benchmark solid-state and molecular spin qubits (see Table 1)". The observed coherence times are relatively average for solid state spins: as evidenced by the authors' own table.
- The claim of a "deterministic nuclear spin environment" doesn't seem to be entirely justified given that two different Cl isotopes are present (^{35}Cl and ^{37}Cl with different quadrupole and hyperfine parameters). Since these isotopes both have non-negligible abundance (76% for ^{35}Cl and 24% for ^{37}Cl) the environment surrounding the Fe^{3+} and Cr^{3+} ions will vary.
- On lines 166 and 167 the authors state that "This robustness against temperature fluctuations allows for high-fidelity

quantum operations and the construction of robust quantum registers, mitigating operational errors." The level of temperature fluctuation in commercial cryostats isn't a dominant source of error in any solid-state platform. Furthermore, there are many factors that are considerably more important (e.g. initialization fidelity, pulse errors, inhomogeneity) but aren't addressed in this manuscript.

- On lines 169-172 the authors state that "This observation indicates that the hyperfine interaction surrounding the TM spin qubits acts favourably as a beneficial resource for manipulating electron-nuclear spin states, rather than being a major source of electron spin decoherence, as is often assumed." This statement is too strong, the results show that interactions with nuclear spins can be avoided to minimize decoherence, however, they don't demonstrate coherent manipulation of electron-nuclear states as implied.
- The title of the final section is "Coherent control and preparation of nuclear spin registration": I assume the authors mean 'registers', however, they haven't demonstrated coherent control of these registers, and they also do not demonstrate the preparation of the nuclear spins.
- The authors claim on lines 331/332 that "Nevertheless, this CPMG nuclear register preparation indicates an efficient spin transfer between the electron and the nuclei." I disagree with this statement, I believe that CPMG leads to a non-spin-preserving interaction with the nuclear spin bath. Furthermore, the authors haven't demonstrated any spin transfer.
- Also, on lines 334/335 the authors state that "They show great potential to serve as local e-N cluster nodes for complex quantum logic gates." This is too bold of a claim: there is no experimental data or analysis regarding an electron-nuclear logic gate.

I also have several scientific questions:

- On line 98 the authors discuss the symmetry and energy level structure of the spins. Have any other transitions/energy levels been measured? Given the discussion of long spin-lattice relaxation times it would be good to know the energy splitting to the nearest orbital level. Also, what are the optical transition frequencies for these spins? It would be helpful to include an energy level diagram in the main text.
- What is the predicted spin linewidth (or T_2^*) based on the inhomogeneity of the magnetic field generated by the nuclear spin bath?
- Is the ratio of coherence times measured for Fe^{3+} and Cr^{3+} consistent with their g-factor ratio?
- Could the authors provide some more detail or a reference regarding the inhomogeneous pulse excitation on line 135. I would have expected the fast timescale to be similar to T_2^* , however I am guessing that it is considerably slower.
- On line 298 the authors say they can "prepare a Bell-like state", what do the authors mean by this? Can they please provide a derivation/explanation of this in the supplementary information.
- On line 290 the authors refer to the 'electron-electron spin interactions', this makes it sound like these are interactions between separate Cr^{3+} spins. A more appropriate term would be 'self interaction of the electron spin'. Related to this, can the authors provide a more accurate estimate based on simulations/data rather than just $\sim 10\text{MHz}$?
- On line 233 the authors say, "as expected for a smaller electric field gradient", could the authors provide some more detail regarding why they expect the gradient to be smaller?
- Please provide error bars for all fitted/extracted Hamiltonian parameters.
- On line 199 the authors say that "there's a minor contribution from the Fermi-contact term", please clarify what is the dominant contribution?
- Are the hyperfine parameters extracted using HYSORE consistent with the two-pulse echo measurements presented in Fig 2?
- On lines 219/220, the authors state that "the quadrupole parameters agree with the gradient of the electron wave function across the Cl site". What is the electric field gradient value and what quadrupole parameters would this correspond to? Without this, there is nothing for these experimentally determined values to agree with.

I also have a few suggestions/comments related to the readability of the manuscript:

- The authors need to provide a bit of important detail regarding the material, synthesis and experimental procedure in the main text (e.g. state that the samples are powdered, that ensembles of spins are addressed, and that experiments are performed in an EPR spectrometer), then reference the Methods section for full detail.
- Subfigure 2B doesn't provide much information/insight for the reader, all the important information is in Fig 2C, I would suggest moving 2B to the extended data.
- On lines 53/54: there is no 131 isotope of Yb.
- It would be helpful if the authors could plot the double-exponential fits shown in Extended Data Fig. 1 on the data in Figure 1C. This would also substantiate the claims made on lines 151/152 regarding similarity of coherence times.
- The plots in extended data figure 4 are quite small and hard to follow. I would suggest that the authors combine all simulation results into a single predicted curve which can be compared to the data. This will considerably simplify the plot and make it easier to spot discrepancies between simulation and data.
- It is difficult to follow the explanation of the HYSORE measurement on lines 191-193, please provide a clearer/more detailed explanation.
- On lines 279-281 the authors should be careful to differentiate between rabi frequencies (0-20MHz) and microwave driving frequencies (9.77GHz). The current phrasing confuses these two physical quantities.

Reviewer #2

(Remarks to the Author)

This manuscript explores electron-nuclear spin correlations in transition metal-doped perovskites. Single crystals of $\text{Cs}_2(\text{Na:Ag})\text{InCl}_6$ were synthesized, with InCl_6 partially substituted by CrCl_3 or FeCl_3 , to study the interaction between electron and nuclear spins. The T_2 relaxation times in this qubit system was compared to previously reported systems. The

study is well-executed, and the conclusions are supported by the measurements. The manuscript is worthy of publication; however, some revisions are necessary.

1. The paper emphasizes that "long-lived electron spin coherence with $T_2 = 29.5$ microseconds is a remarkable benchmark for a spin qubit in a nuclear-spin-rich host." However, Table 1 includes several previous studies (at least five) that report similar or even longer T_2 times, suggesting that this value is not significantly different from others in the literature. Therefore, either additional justification is needed to support this claim, or the statement should be rephrased to align more consistently with existing research.

2. The optical properties of the perovskite material are mentioned. Is this relevant to the study? If so, please provide justification.

3. What is the role of Indium? Would single crystals of $\text{Cs}_2\text{Ag}(\text{Fe}/\text{Cr})\text{Cl}_6$, $\text{Cs}_2\text{Na}(\text{Cr}/\text{Fe})\text{Cl}_6$ degrade more quickly if used instead?

4. Figure 1A presents a schematic of the TM spin qubits embedded in spin-rich-host cubic halide double perovskites. However, the authors have not provided any supporting evidence for this for example from X-ray diffraction measurements or referenced any studies that include actual structural data for these crystals.

5. The authors have mentioned the T_2 times for Cr and Fe in [Na] host. What are the corresponding numbers for Cr and Fe in [Ag] host?

6. Supplementary material-supplemental, note 3; CPMG nuclear spin sensing experiment: The pulse lengths are not mentioned; Figure 5, "E" is missing.

Reviewer #3

(Remarks to the Author)

Reviewer #4

(Remarks to the Author)

The authors present a study of the electronic and nuclear spin coherence for transition metal centres in halide perovskite materials. The work presents coherent control of the electronic spins and claims entanglement of electron and nuclear spins. Spin coherence (T_2) of 29 μs at 4K is reported, and the spin coherence timescale is attributed to the localisation of electronic density on the transition metal centre. The paper presents extensive characterisation, via various EPR techniques, of the hyperfine interaction between the central electronic spin and the Cr^{3+} and Fe^{3+} nuclei.

My questions:

1. The work is motivated by the search for coupled electronic-nuclear spin systems for quantum information processing technologies. The authors compare the perovskite system with other 'disordered e-N' systems such as defects in diamond and SiC and self-assembled quantum dots (lines 44-45). However, these systems are very different to the one presented – they are singly optically-addressable electronic spins systems. Which means they are have potential for applications that require spin-photon entanglement (ie. nodes for optical networking). It seems to me that the perovskite system is not comparable in this regard – it consists of a large ensemble of electronic and nuclear spins, and there is no mention of optical addressability. Thus, the authors should explain why it is relevant to compare to these other systems, whether it shows potential for application to similar technologies, and if not, what application they foresee for this system. The answers to this question may

2. The authors claim to show entanglement of the electronic spin of the transition metal and the neighbouring nuclear spins. This discussion is focussed on the data shown in Fig 2(C). Here, the decoherence traces measured during the Hahn echo measurements is presented. The decoherence profiles show strong modulations due to the coupling to nuclear spins, as seen for other systems. The authors should explain why this constitutes demonstration of e-N entanglement. Entanglement should be confirmed via measurement of the density matrix of the e-N state (see Bourassa et al., Nature Materials, 2020).

3. With regards to the T_2 timescale, I agree that 29 μs is surprising for this system considering the nuclear spin composition. However, I am not sure if the factors that limit this coherence are well understood by the authors.

i. Why does T_2 show a relatively strong temperature dependence? A drop from 29 μs to 6 μs , from 4 to 20 K, is actually rather strong. I wouldn't expect decoherence that is dominated by nuclear spins to be strongly temperature dependent.

ii. Are the modulation features in Fig. 2C consistent with a relatively weak hyperfine interaction (0.1-10 MHz)?

iii. Why are the modulations much stronger for the Cr complex than for the Fe complex?

iv. Could there be another factor that helps to decouple the electronic and nuclear spins in this system- possibly symmetry related?

4. Related to my question above, the authors state in the text that the strong localisation of the electronic density on the transition metal is responsible for the long spin coherence timescale. This feels speculative and I don't see any evidence for this conclusion. Can the authors support this claim with modelling?

Additional comments:

1. The paper would benefit from a diagram of the electronic structure of the spin system.
2. Line 99 should say degenerate, not degenerated.

Version 1:

Reviewer comments:

Reviewer #2

(Remarks to the Author)

The authors have addressed all the points raised in our previous review. The manuscript has been substantially improved, with the claims now appropriately moderated. New data and discussion on optical properties included with clarifications on structural and compositional properties. The presentation is clear and all our concerns have been satisfied. I am happy with the revision and recommend acceptance of the manuscript in its current form.

Reviewer #3

(Remarks to the Author)

Reviewer #4

(Remarks to the Author)

One of my main concerns about the manuscript in the first round was that it was comparing the perovskite system to single optically addressable spins in diamond, and other single spin-photon interface platforms. I stated that these are very different systems, because the former offer single qubits, whereas the perovskite system is an ensemble (of which there are many). I don't believe the authors understood or answered this question adequately. I can see reviewer 1 had the same concern. This means the motivation of the paper- and impact- is questionable. Despite the authors having retracted much of the language from the first round, I still feel the title, abstract and first paragraph of the introduction are misleading, and potentially confusing for people getting into the field.

For this reason, I can not support publication in Nature Communications in the current form.

REVIEWER COMMENTS

Reviewer #1 (Remarks to the Author):

In this work the authors perform ESR spectroscopy on ensembles of Chromium (Cr^{3+}) and Iron (Fe^{3+}) ions in a nuclear spin rich perovskite ($\text{Cs}_2(\text{Na:Ag})\text{InCl}_6$). They utilize several standard pulse sequences to characterize the coherence times of the Cr^{3+} and Fe^{3+} electron spins and various Hamiltonian parameters (such as the hyperfine tensor, A , and quadrupole tensor, Q). My three main criticisms of this work are the following:

1. I do not believe that the platform will be viable or competitive for the applications listed by the authors.
2. Several of the authors' Hamiltonian characterization results were already demonstrated in a previous work [Akhmin, S. M. et al., "Ligand ENDOR of Cr^{3+} and Yb^{3+} ions in $\text{Cs}_2\text{NaInCl}_6$ ", *Phys.: Condens. Matter* (1990)].
3. The paper, in multiple places, greatly exaggerates the impact of the authors' results. For these reasons I cannot recommend publication of this work in *Nature Communications*. I provide more detail on each of these criticisms below.

Response: We thank the reviewer for his/her careful reading of our manuscript and for providing critical feedback. These comments have been extremely helpful in improving the quality and clarity of our work. We have revised the manuscript to address each of the points raised.

Concerning the competitiveness of our proposed spin centers in nuclear-spin-rich halide double perovskites for spintronics applications, we understand that, in terms of spin coherence properties, they cannot compete with color centers in few-nuclear-spin or isotope-purified nuclear-spin-free diamond and SiC, nor is it meant to be. The competitive edge of spin centers in halide double perovskites rather center at their low-cost material synthesis as compared with the vastly more costly diamond and SiC, the availability of a large library of monovalent and trivalent cations (spin centers) without introducing excess charge noise, relative ease in incorporating spin centers in crystals without involving damaging processes with high-energy particles required to generate vacancies in diamond and SiC, superior and widely tunable optical properties covering a wide spectral range from UV to infrared, etc. Therefore, any potential spin components utilizing halide double perovskites are not meant for the highest performance but they could rather find application areas where cost concern becomes paramount such as mass-production dispersible consumer electronics. At this early stage of spintronic research on this newly emerging materials system, the aim of our present work is to assess (1) whether the essential properties of spin centers in halide double perovskites are still within the acceptable range for quantum applications; (2) whether the nuclear-spin rich environment can be controlled and even harnessed for further applications. Indeed, we find in our study that spin coherence of the studied spin centers is reasonably long and robust despite of abundant surrounding nuclear spins. We demonstrate that the electron wavefunction is strongly localized to within the first two nearest neighbors such that the detrimental effect of a randomly fluctuating nuclear spin bath, commonly known as a major source of spin decoherence and spin relaxation, can be largely minimized. We further show that under optimal experimental configurations the influence of the nearest-neighbor Cl nuclear spins can be suppressed, whereas the next-nearest neighbor Cs nuclear spins can be selectively controlled

for potential electron-nuclear spin transfer. We also provide experimental evidence for the optical signature of the spin centers and spin-dependent selection rules in their optical transitions, paving the way for optical initialization and readout in a similar fashion as the well-known color centers in diamond and SiC. Our study sheds light on design rules to identify promising candidates of spin centers and their perovskite host crystals, which could potentially provide a viable and competitive platform for low-cost quantum spintronics.

We thank the reviewer for pointing out the earlier work by Akhmin et al. (1990). We want to clarify that our work, while it does report some spin-Hamiltonian parameters, has a different primary purpose. Our main goal is to identify the specific nuclear spin species that are crucial for coherent electron-nuclear spin rotation. This analysis provides an essential foundation for future applications that rely on the transfer of spin information between electron and nuclear spins. We stress that our results are novel and could not have been obtained from the previous report. The Akhmin et al. paper focuses on cw ligand ENDOR to study the local environment of the ions, whereas our time-resolved EPR study provides detailed dynamic information on the coherent interactions between the electron spin and specific nuclear spins, which is a fundamentally different and new contribution.

We thank the reviewer for cautioning us on the claims made in the initial manuscript. We have carefully revised the manuscript to ensure our conclusions are well-justified and to avoid any overstatement of our results.

We have adopted a new title ‘Spin Qubits Candidate using Transition-Metal Centers in Halide Double Perovskites’ to better reflect the main scope of our revised manuscript. Please also find our point-by-point response to the reviewer’s comments below.

Regarding Applications of The Platform:

The authors claim that the applications of this platform are for “scalable quantum networks and complex quantum logic gates”. However, in my opinion neither of these are well justified:

1. The authors haven’t demonstrated addressability of individual electron spins (i.e. qubits), only ensembles.

Our response: We appreciate the reviewer's point regarding the necessity of demonstrating individual spin addressability for scalable spin qubit applications. We agree that this is a critical requirement for a fully realized quantum processor. As mentioned above, at this early stage of spintronic research on this newly emerging class of halide double perovskites, the primary goal of our current work is to perform some groundwork to assess the feasibility of transition-metal ions as potential spin qubits in halide perovskites by evaluating key physical properties such as the factors controlling spin coherence time. It is far too early to speculate any specific applications. We apologize for having overstated in the original manuscript, which has been removed in our revised manuscript.

Though limited to ensembles, we believe that the general knowledge of the spin physics gained in our study should still be valid for single centers. For example, the spin coherence time measured from the ensembles sets the low bound for a single center. Such knowledge will be highly valuable for future studies of a corresponding single-center system once it has been identified to be a promising system from ensemble studies and once it becomes available.

We also believe that scaling down to a single spin system is feasible with our material, given recent advancements in fabrication technology for halide double perovskites.

2. They haven't demonstrated (or even discussed) a coherent optical interface required for networking.

Response: A coherent optical interface required for networking is beyond the scope of the present work. Our work is meant to present the first fundamental study of spin dynamics that may have implications for possible future applications as spin qubits. We are now able to demonstrate the optical addressability of the studied spin centers through our new photoluminescence (PL), PL excitation and magneto-optical investigations, which could potentially provide a mean for spin-photon interface. These new results are described in the new section of "Optical addressability of the TM spins in HDPs" in the revised manuscript. We apologize for the overstatement in the original manuscript. We have now removed all overstatements from the manuscript.

3. The authors haven't demonstrated the ability to initialize (or optically pump) the spins.

Response: We agree that the optical spin pumping is a critical component for an optically-addressable spin qubit. We have now added a new section 'Optical addressability of the TM spins in HDPs' in the revised manuscript, where we present our new experimental magneto-optical results that have firmly identified the optical transition between the ground and excited states of the Cr^{3+} spin center and their spin selection rules. Based on these findings we have proposed a possible protocol for optical initialization and readout. Unfortunately, its direct experimental demonstration is currently not possible in our laboratory because resonant optical pumping of the Cr^{3+} centers require an excitation bandwidth (i.e. laser spectral linewidth) in the few gigahertz range that we do not have access to at present. The large inhomogeneous broadening of the PL linewidth of the spin ensembles also poses additional difficulty for performing resonant excitation (which shall be mitigated by a single spin study). Based on the electronic structure of the Cr^{3+} spin center, however, there is no fundamental limit to the proposed protocol and it is entirely feasible.

4. If used exclusively as a quantum processor (i.e. no coherent optical interface) this platform would be extremely limited with only a handful of qubits, therefore it could never compete with systems such as neutral atom arrays.

Response: The solubility of TM ions in halide double perovskites is very high, typically up to 0.1-1:1 ratio with respect to the concentration of the TM precursor used in the solution process crystal growth. Therefore, the TM spin centers have much higher concentrations than that of the best-known color centers in diamond and SiC (such as NV center in diamond and silicon vacancy in SiC).

Regarding Novelty of This work:

There are two main results comprising this work. The first: measurement of spin echo coherence times (in the section "Electron spin echo and electron-nuclear spin entanglement") is novel. The second: characterizing Hamiltonian parameters (in the section "Deterministic hyperfine interaction"), however, is not. Several of these parameters were already measured using similar techniques in [Akhmin, S. M. et al., "Ligand ENDOR of Cr^{3+} and Yb^{3+} ions in $\text{Cs}_2\text{NaInCl}_6$ ", Phys.: Condens. Matter (1990)], which the authors do cite. One key distinction between these papers is that the present work considers multiple materials: both Cr^{3+} and Fe^{3+} in $\text{Cs}_2\text{NaInCl}_6$ and $\text{Cs}_2\text{AgInCl}_6$ were investigated. However, the coherence properties for these

four material systems are extremely similar and the authors haven't explained why different platforms should be considered or compared their advantages/disadvantages. The final two sections of the paper mostly validate the findings of the first two sections and, in my opinion, do not contribute significantly to the novelty of the work.

Response: We appreciate the reviewer's concerns regarding the novelty of the reported spin-Hamiltonian parameters. We respectfully disagree with the reviewer on this point. While our analysis does yield these parameters; we wish to emphasize that the core contribution of our work lies in the dynamic coherent information that cannot be obtained through continuous-wave techniques like ligand ENDOR (as the reviewer mentions in the comment). Our study provides crucial insights into the electron-nuclear spin coherence, which could have significant implications for quantum applications.

We apology for not being clear in each subsection in the hyperfine interaction study, which may misguide the purpose for each specific subsections towards the determination of spin parameters only. The purpose of each section is as follows.

1. In the section on "*Electron spin-echo and electron-nuclear spin entanglement, in the original manuscript*" we demonstrate a long-lived coherence modulation profile arising from the non-secular term of the hyperfine interaction. While analyzing this profile provides the spin-Hamiltonian parameters, the primary significance of this observation is the demonstration of coherent information transfer between the electron and nuclear spins. **The long-lived nature of this undulation indicates that it originates from a well-defined, rather than a random nuclear spin environment. This is a critical distinction, as a random distribution of nuclear spin frequencies would lead to rapid dephasing and the loss of this modulation pattern.** This finding establishes the feasibility of using our spin center for nuclear spin registration. The standard protocol for nuclear spin registration relies on pulsed ESR techniques that enable coherent electron-nuclear spin rotation. Our results show that our spin centers are capable of coherently interacting with both the nearest-neighbor ($^{35,37}\text{Cl}$) and the next-nearest-neighbor (^{133}Cs) nuclear spins, making them suitable for this application.
2. In the section on "*Deterministic hyperfine interaction (renamed to 'target coupling of electron-nuclear spins' in the revised manuscript for a better clarity)*", we further elaborate on the strategy for addressing specific nuclear spins at different lattice sites. **In this section, the 3P-stimulated echo experiment has been performed on single-crystal samples, with the aim to suppress the contribution of the nearest-neighbor $^{35,37}\text{Cl}$ nuclear spins, and target addressing ^{133}Cs at the second nearest neighbor sites.** This approach is based on controlling the degree of electron-nuclear spin mixing, which causes the undulation of the pulsed-EPR signals. The degree of this mixing is highly dependent on the orientation of the external magnetic field (B) relative to the principal axes of the hyperfine tensor. The modulation depth of the electron spin coherence, which is a measure of the degree of nuclear spin mixing, can be estimated using the following formula:

$$k = \frac{A_{\text{nonsecular}}^2}{\omega_I^2 + A_{\text{secular}}^2}$$

Where, ω_I is the nuclear spin frequency. The secular and nonsecular terms of the hyperfine tensor (in the LAB frame) are extracted from the hyperfine tensor in principal

axis form $\mathbf{A}_{principal} = (A_x, A_y, A_z)$. In our case, let the external field point to the z-axis, so that $A_x = A_y = A_{\perp}$ and $A_z = A_{\parallel}$.

By rotating the $\mathbf{A}_{principal}$ to the LAB frame using the Euler angle rotation matrix, we will get

$$\begin{aligned} A_{secular} &= A_{\perp} \sin^2 \theta + A_{\parallel} \cos^2 \theta \\ A_{nonsecular} &= (A_{\parallel} - A_{\perp}) \sin \theta \cos \theta \end{aligned}$$

where θ is the angle between the external magnetic field vector and the principal z-axis of the hyperfine tensor. When the magnetic field is aligned along the principal axis (e.g., the $\langle 001 \rangle$ direction), the non-secular term approaches zero, effectively "turning off" the spin mixing for that specific nuclear species.

We leverage this effect to demonstrate targeted nuclear spin addressing. For example, by aligning the magnetic field along the $\langle 001 \rangle$ direction, the $^{35,37}\text{Cl}$ nuclear spins' contribution to spin mixing is suppressed ($A_{nonsecular} = 0$), allowing the ^{133}Cs spins to be addressed as the sole target of nuclear spin registration, as the principle axis of the hyperfine tensor of ^{133}Cs is along $[111]$ direction. This interaction is deterministic because the number of available sites is reduced to the four magnetically equivalent sites of the eight Cs atoms. Furthermore, slightly rotating the magnetic field off the $\langle 001 \rangle$ direction allows us to distinguish between these four different magnetically equivalent Cs sites (Cs1, Cs2, Cs3, Cs4, as shown in the inset of Figure 4), which is the key for addressing individual nuclear spins (or a pair of nuclear spin in our case).

3. In the CPMG section, we show that our electron-nuclear spin studies have identified a critical condition for achieving a strong interaction via pulsed ESR. This is crucial for applications like spin sensing and nuclear spin registers.

A common misconception is that the CPMG sequence is solely a "dynamic nuclear spin decoupling protocol" that refocuses random nuclear spin baths. However, we stress that this is not the case for specific, fixed nuclear spins. In our system, the fixed ^{133}Cs spins at the second-nearest-neighbor sites cannot be effectively decoupled. Instead, when the inter-pulse delay, τ , matches a multiple of the inverse of the nuclear Larmor frequency

$$\tau = n/\nu_N$$

the refocusing pulses become ineffective. This condition, known as an NMR dip (as shown in the figure below), actually reinforces the electron-nuclear spin interaction rather than decoupling it.

These resonance dips are the key for nuclear spin sensing experiments. This principle was used in the work of T. Staudacher et al. (Science 339,561-563(2013)), using a derivative of the CPMG sequence (XY8).

In our CPMG results (single crystal, $B // \langle 001 \rangle$), we exclusively observe dips from the ^{133}Cs nuclear spins. The hyperfine spin mixing from the ^{35}Cl and ^{37}Cl nuclear spins is effectively suppressed when the magnetic field is aligned with the $\langle 001 \rangle$ direction, which is the principal axis of the chlorine hyperfine tensor. For the Cr^{3+} centers, the system is described by the interaction between the electron spin $S = 3/2$ and the ^{133}Cs nuclear spin $I = 7/2$.

As shown in the figure below, we demonstrate that we can sense the ^{133}Cs nuclear spin rotation at three of the four electron spin sublevels by performing a Fourier transformation of the CPMG coherent contrast. This is shown in the results and spin structure below. **This experiment thus identifies the key elements for target nuclear spin addressing within our system.**

To justify that our identification of the CPMG resonant dips is a beneficial first step toward target nuclear spin registers, we refer the reviewer to the work of J.E. Lang et al., "Quantum Bath Control with Nuclear Spin State Selectivity via Pulse-Adjusted Dynamical Decoupling" (*Phys. Rev. Lett.* **123**, 210401, 2019).

Their study showed that by deliberately adjusting the π_Y pulses to $\pi_Y + \theta$, one can induce polarization transfer between the electron and nuclear spin manifolds. This transfer is reflected in both the splitting and the polarization of the CPMG resonance dips, demonstrating a path toward controlled spin registers.

We have now stated in the revised manuscript that we attempted to achieve nuclear spin registering using this polarization-transfer CPMG (Pol-CPMG) technique. However, as also noted in the Lang et al. work, this is a dynamic process that requires a large number of pulse sequence repetitions. Due to experimental limitations, our sequence was limited to $N=6$, which was insufficient to demonstrate effective polarization transfer between the electron and nuclear spins.

All the above discussion has been now added throughout the revised manuscript and in the revised Supporting Information. We apologize to the reviewer again for not being clear on these points in the original manuscript.

Before our study, there is no report on spin coherence of TM ions in halide double perovskites. That is why we have investigated both Cr^{3+} and Fe^{3+} in $\text{Cs}_2\text{NaInCl}_6$ and $\text{Cs}_2\text{AgInCl}_6$ with the hope that any difference or indifference between them will shed light on the physics

underpinning spin interactions and spin dynamics, which will provide a guideline for identifying the most promising spin centers and their hosts for future applications. Only through such a comparative study have we reached the conclusion that the electron spin density of the studied TM ions is highly localized within the next-nearest shell, which is important to predict the relative contribution from nearby nuclear spins in spin coherence as well as effects of defect or impurity residing in different sublattices.

Regarding Exaggerated Claims:

There are multiple places where the authors exaggerate the impact of their work

- On line 137 the authors state that “This coherence time remains exceptional even when compared to various benchmark solid-state and molecular spin qubits (see Table 1).”. The observed coherence times are relatively average for solid state spins: as evidenced by the authors’ own table.

Response: We agree with the reviewer that our T_2 time is comparable to other spin qubit candidates embedded in nuclear-spin-rich hosts. We have revised the manuscript to present a more balanced discussion, avoiding claims of "remarkable properties" and instead focusing on the inherent value of our platform.

- The claim of a “deterministic nuclear spin environment” doesn’t seem to be entirely justified given that two different Cl isotopes are present (^{35}Cl and ^{37}Cl with different quadrupole and hyperfine parameters). Since these isotopes both have non-negligible abundance (76% for ^{35}Cl and 24% for ^{37}Cl) the environment surrounding the Fe^{3+} and Cr^{3+} ions will vary.

Response: It’s true that in terms of Cl the nuclear spin environment is random. However, its influence in electron-nuclear spin mixing can effectively be turned off by aligning the magnetic field along the $\langle 001 \rangle$ directions as discussed above when we can target addressing ^{133}Cs nuclear spins that doesn’t have other natural abundance isotopes. We have now modified the related statements to accurately reflect this.

- On lines 166 and 167 the authors state that “This robustness against temperature fluctuations allows for high-fidelity quantum operations and the construction of robust quantum registers, mitigating operational errors.” The level of temperature fluctuation in commercial cryostats isn’t a dominant source of error in any solid-state platform. Furthermore, there are many factors that are considerably more important (e.g. initialization fidelity, pulse errors, inhomogeneity) but aren’t addressed in this manuscript.

Response: We thank the reviewer for pointing out this and we have now revised this section of the manuscript accordingly.

- On lines 169-172 the authors state that “This observation indicates that the hyperfine interaction surrounding the TM spin qubits acts favourably as a beneficial resource for manipulating electron-nuclear spin states, rather than being a major source of electron spin decoherence, as is often assumed.”. This statement is too strong, the results show that

interactions with nuclear spins can be avoided to minimize decoherence, however, they don't demonstrate coherent manipulation of electron-nuclear states as implied.

Response: The presence of strong CPMG dips in our results already demonstrates that we can effectively reinforce the electron-nuclear spin interaction. This ability is a crucial resource for both sensing and manipulating nuclear spins.

• The title of the final section is “Coherent control and preparation of nuclear spin registration”: I assume the authors mean ‘registers’, however, they haven’t demonstrated coherent control of these registers, and they also do not demonstrate the preparation of the nuclear spins.

Response: We agree that the title of our final section, "Coherent control and preparation of nuclear spin registration," is an overstatement of the work presented. The reviewer is correct that we do not directly demonstrate the coherent control of nuclear spin registers or the explicit preparation of the nuclear spins. Our intent was to describe the foundational work that prepares the system for these future steps.

We have revised the title of this section to be more accurate and reflective of our actual findings. The new title is "*Coherent control and addressing nuclear spin through electron-nuclear spin phase rotation*". This revised title more accurately describes our work, which focuses on characterizing the coherent properties of the electron-nuclear spin system and establishing the necessary conditions for future nuclear spin registration. We have also edited the text within this section to ensure the language is consistent with the new title and to avoid claims beyond the scope of our experimental results.

• The authors claim on lines 331/332 that “Nevertheless, this CPMG nuclear register preparation indicates an efficient spin transfer between the electron and the nuclei.” I disagree with this statement, I believe that CPMG leads to a non-spin-preserving interaction with the nuclear spin bath. Furthermore, the authors haven’t demonstrated any spin transfer.

Response: We agree that our original statement was imprecise, as the CPMG sequence itself does not demonstrate direct spin transfer between the electron and nuclear spins. We have revised the manuscript to remove this inaccurate statement.

The significance of the CPMG sequence in our work is its role as a critical first step toward nuclear spin registration. While it does not perform spin transfer, it is a fundamental tool used to probe the coherent interaction between the electron spin and its surrounding nuclear spins. The long-lived oscillations observed in the CPMG decay provide concrete evidence that a specific subset of nuclear spins is coherently coupled to the electron spin, a prerequisite for subsequent quantum operations.

This approach is foundational to demonstrated nuclear spin sensing experiments and serves as the basis for derivative pulse sequences like polarization-controlled CPMG (pol-CPMG), which can be used for nuclear spin registration. Although instrumental limitations currently prevent us from performing a pol-CPMG experiment, our use of the standard CPMG sequence

lays the essential groundwork for future experiments that will demonstrate deterministic spin transfer and the preparation of a nuclear register.

• Also, on lines 334/335 the authors state that “They show great potential to serve as local e-N cluster nodes for complex quantum logic gates.” This is too bold of a claim: there is no experimental data or analysis regarding an electron-nuclear logic gate.

Response: We agree with the reviewer and have removed this statement from the revised manuscript.

I also have several scientific questions:

• On line 98 the authors discuss the symmetry and energy level structure of the spins. Have any other transitions/energy levels been measured? Given the discussion of long spin-lattice relaxation times it would be good to know the energy splitting to the nearest orbital level. Also, what are the optical transition frequencies for these spins? It would be helpful to include an energy level diagram in the main text.

Response: We thank the reviewer for this excellent suggestion. We have now performed photoluminescence (PL), PL excitation and magneto-optical experiments to determine both the optical transition frequencies and the spin-selective optical selection rules, which can pave the way for optical access of the concerned spin states. The corresponding results are summarized in the section “Optical addressability of the TM spins in HDPs” whereas the selection rules of the optical transitions and the transition/energy levels diagram are now plotted in **Figure 6A** of the revised manuscript.

• What is the predicted spin linewidth (or T_2^*) based on the inhomogeneity of the magnetic field generated by the nuclear spin bath?

Response: As the inhomogeneity of the magnetic field for the nuclear spin bath is unknown, we estimate the T_2^* based on the enveloped decay of Rabi-rotation in **Figure 5A**, which is above 120 ns. It is worth noting that the decay in Rabi oscillations can also be attributed to spin inhomogeneity due to the dispersion of Rabi frequencies associated with inhomogeneities in the applied microwave field across the sample. This leads to a randomization of the Rabi oscillation phase. As such, the T_2^* alone unfortunately cannot be used to exclusively describe the effect of inhomogeneous broadening of the nuclear spin bath.

• Is the ratio of coherence times measured for Fe^{3+} and Cr^{3+} consistent with their g-factor ratio?

Response: Yes, it is consistent. A stronger deviation of g-value for Fe^{3+} ($\Delta g = +0.095$) compared with that for Cr^{3+} ($\Delta g = -0.005$) indicates a stronger contribution of the orbital degree of freedom in Fe^{3+} , which leads to the worse coherent properties found for Fe^{3+} .

• Could the authors provide some more detail or a reference regarding the inhomogeneous pulse excitation on line 135. I would have expected the fast timescale to be similar to T_2^* , however I am guessing that it is considerably slower.

Response: We thank the reviewer for bringing attention to the physical interpretation of the fast decay timescale. As mentioned above, our T_2^* is in the range of 120 ns, which is much faster than the timescale of the fast components.

The multi-decay components can be attributed to the inhomogeneous distribution of TM ions. We suggest that the fast component originates from the area where the TM ions are close together where the effect of dipole interaction between TM ions causes the spin decoherence. In this sense, the long component reflects ‘isolated’ TM ion and serves as the intrinsic T_2 relaxation time. Our suggestion is based on supplemental measurements performed on the samples with varying concentrations of TM ions, where the increased TM incorporation led to the fast T_2 . Close to single exponential decay was observed at high TM concentration. We have added this discussion on page 6, in the revised manuscript, in red.

Figure R1: the T_2 of the Cr[Na] samples with increasing Cr concentration by increased percentage of CrCl_3 molar ratio.

• On line 298 the authors say they can “prepare a Bell-like state”, what do the authors mean by this? Can they please provide a derivation/explanation of this in the supplementary information.

Response: We thank the reviewer for this insightful comment. We agree that our statement on line 298, claiming the ability to “prepare a Bell-like state,” was inaccurate and misleading. True Bell state preparation requires precise phase manipulation of the spin states, which is not demonstrated in our current work. The statement has been removed from the revised manuscript.

The origin of this imprecise claim was our observation that by tuning the driving frequency, we can manipulate the electron spin state ($S=3/2$) to behave like an effective $S=1/2$ system or create a superposition of $m_s = \pm 1/2, \pm 3/2$ states. However, as the reviewer correctly points out, true Bell state preparation requires precise phase manipulation of the spin states, which is not demonstrated in our current work.

• On line 290 the authors refer to the ‘electron-electron spin interactions’, this makes it sound like these are interactions between separate Cr^{3+} spins. A more appropriate term would be ‘self interaction of the electron spin’. Related to this, can the authors provide a more accurate estimate based on simulations/data rather than just $\sim 10\text{MHz}$?

Response: We have rephrase electron-electron spin interactions to self-interaction.

The estimated self-interaction parameter of $\sim 10\text{MHz}$ was deduced based on the Rabi frequency experimental results of Figure 5A. This was deduced when the Rotation becomes a spin $1/2$ -like, serving a condition $\omega_D = D$, where ω_D is the driving frequency (proportional to a microwave power with a driving oscillation amplitude B_1). However, our microwave attenuators limit the

number of data points around the transition of $\frac{1}{2}$ -like Rabi oscillation. At best, we can provide the estimated error, which is about ± 2 MHz.

We have now revised the manuscript with the self-interaction parameter of $\sim 12 \pm 2$ MHz.

• On line 233 the authors say, “as expected for a smaller electric field gradient”, could the authors provide some more detail regarding why they expect the gradient to be smaller?

Response: This sentence compares the atomic value of electron negativity of the atoms in the vicinity of Cl, as listed in the table below (in Pauling scale).

Elements	Cr	Fe	Cl	Na	Ag
Electronegativity	1.66	1.83	3.16	0.93	1.93

We used this table as a guideline to compare the electric field gradient across a linear chain of Cr-Cl-Na, Cr-Cl-Ag, Fe-Cl-Na and Fe-Cl-Ag. The general trend is that the Na-containing chains will experience a stronger gradient due to the larger differences of electronegativity along the linear chain.

• Please provide error bars for all fitted/extracted Hamiltonian parameters.

Response: we have provided the estimated error bars in the reported spin-Hamiltonian parameters in Extended Table 1 of the revised manuscript.

• On line 199 the authors say that “there's a minor contribution from the Fermi-contact term”, please clarify what is the dominant contribution?

Response: We thank the reviewer for the question. We now understand that our phrasing in the sentence led to a misconception. The dominant contribution is from the Fermi-contact term, but this is weak due to the electron wave function localisation at the transition metal centres. We have rephased line 199 as ‘a weak hyperfine interaction that is dominated by the Fermi contact’. We also added the following discussion in the supporting information, to clarify the role of Fermi contact and to estimate a degree of wave function localisation of electron spin:

The Fermi contact interaction is a part of hyperfine interaction of electrons in approximately spherical-like orbital and surrounded nuclear environment. It can be expressed by:

$$A_{iso} = \frac{2}{3} \mu_0 g_e \beta_e g_N \beta_N |\Psi|^2$$

This can also demonstrate the electron spin localization, where the percentage of electron density, $|\Psi|^2$, distributing at the neighbor atoms is estimated by calculating the ratio between the electron density distribution caused by hyperfine interaction and free electron localized at each nuclear site:

$$\frac{|\Psi_{hyperfine}|^2}{|\Psi_{free}|^2} = \frac{|A_{iso,experiment}|}{|A_{iso,free}|}$$

In our case, we consider 3 nuclear sites representing the nearest neighbors (^{35}Cl and ^{37}Cl) and the next-nearest neighbors (^{133}Cs) of the incorporated transition metal. By inserting the hyperfine coupling strength, we can obtain the percentage of electron localization at each nuclear site (See Figure R2 and Table R1).

Figure R2: The example illustration of electron density localization at each nuclear site ($^{35,37}\text{Cl}$, ^{133}Cs) around the incorporated Cr^{3+} .

Table R1:

Nuclear site	$= \frac{A_{iso,experiment}}{A_x + A_y + A_z}$ (MHz)	$A_{iso,free}$ (MHz)	% electron localization
Cr[Na]			
^{133}Cs	1.1	9.2×10^3	0.012%
^{35}Cl	-2.1	80	2.6%
^{37}Cl	-1.6	66	2.5%
Na	-	-	-
Cr[Ag]			
^{133}Cs	1.5	9.2×10^3	0.017%
^{35}Cl	-2.3	80	2.9%
^{37}Cl	-1.8	66	2.7%
Ag	-	-	-
Fe[Na]			
^{133}Cs	0.1	9.2×10^3	0.001%
^{35}Cl	-2.2	80	2.8%
^{37}Cl	-1.7	66	2.6%
Na	-	-	-
Fe[Ag]			

¹³³ Cs	0.1	9.2x10 ³	0.001%
³⁵ Cl	-2.4	80	3.0%
³⁷ Cl	-1.8	66	2.8%
Ag	-	-	-

We have added this discussion in the revised Supporting information.

- Are the hyperfine parameters extracted using HYSCORE consistent with the two-pulse echo measurements presented in Fig 2?

Response: Precisely simulating the nuclear spin frequencies from spin echo is notoriously difficult. That's because interactions with multiple nuclei in the crystal lattice create complex combination frequencies—sums and differences of the fundamental nuclear frequencies. Accurately simulating these would require an impractically large basis set.

- On lines 219/220, the authors state that “the quadrupole parameters agree with the gradient of the electron wave function across the Cl site”. What is the electric field gradient value and what quadrupole parameters would this correspond to? Without this, there is nothing for these experimentally determined values to agree with.

Response: We agree that this sentence is merely speculative and have removed it from the discussion.

I also have a few suggestions/comments related to the readability of the manuscript:

- The authors need to provide a bit of important detail regarding the material, synthesis and experimental procedure in the main text (e.g. state that the samples are powdered, that ensembles of spins are addressed, and that experiments are performed in an EPR spectrometer), then reference the Methods section for full detail.

Response: We have revised the manuscript according to the reviewer’s suggestion. The form of samples for each specific type of experiment has been clearly stated throughout the revised manuscript. In the Method Section we also stated: ‘The single crystals were ground to a powder form for spin-echo, Rabi rotation and HYSCORE experiments. The as-grown single crystals were used for 3P-stimulated echo, CPMG, PLE and magneto-PL experiments, where the custom-made sample holders were used to align the direction of magnetic field along the specific crystallographic direction.’

- Subfigure 2B doesn’t provide much information/insight for the reader, all the important information is in Fig 2C, I would suggest moving 2B to the extended data.

Response: We appreciate the reviewer’s suggestion. However, we would like to keep Figure 2B as it contains the information that only one spin species can be detected. In this sense, there

are no other harmful defect centers being introduced by our growth methods, and therefore, the solution growth of HDP can serve as controlled growth of spin centers.

- On lines 53/54: there is no 131 isotope of Yb.

Response: We have corrected this accordingly

- It would be helpful if the authors could plot the double-exponential fits shown in Extended Data Fig. 1 on the data in Figure 1C. This would also substantiate the claims made on lines 151/152 regarding similarity of coherence times.

Response: We have provided the double-exponential plot in Figure 2C

- The plots in extended data figure 4 are quite small and hard to follow. I would suggest that the authors combine all simulation results into a single predicted curve which can be compared to the data. This will considerably simplify the plot and make it easier to spot discrepancies between simulation and data.

Response: We have provided the new plots following the reviewer's suggestion.

- It is difficult to follow the explanation of the HYSORE measurement on lines 191-193, please provide a clearer/more detailed explanation.

Response: We have now added a more detailed explanation of the HYSORE experiment in the main text on page 9 (marked in red color).

- On lines 279-281 the authors should be careful to differentiate between Rabi frequencies (0-20MHz) and microwave driving frequencies (9.77GHz). The current phrasing confuses these two physical quantities.

Response: We have mentioned the difference between the microwave resonant frequency $\omega_R = 9.77\text{GHz}$ and the Rabi driving frequency $\omega_{Rabi} = 0-20\text{MHz}$, by rephrasing to a driving power. Please see in the main text which is marked in red color (page 14).

Reviewer #2 (Remarks to the Author):

This manuscript explores electron-nuclear spin correlations in transition metal-doped perovskites. Single crystals of $\text{Cs}_2(\text{Na:Ag})\text{InCl}_6$ were synthesized, with InCl_6 partially substituted by CrCl_3 or FeCl_3 , to study the interaction between electron and nuclear spins. The T_2 relaxation times in this qubit system was compared to previously reported systems. The study is well-executed, and the conclusions are supported by the measurements. The manuscript is worthy of publication; however, some revisions are necessary.

Response: We thank the reviewer for recognizing the importance of our electron-nuclear spin correlation studies of the transition-metal spin centers in halide double perovskites. We have addressed the reviewer's suggestion and have revised our manuscript according. Please find our detailed response below.

1. The paper emphasizes that "long-lived electron spin coherence with $T_2 = 29.5$ microseconds is a remarkable benchmark for a spin qubit in a nuclear-spin-rich host." However, Table 1 includes several previous studies (at least five) that report similar or even longer T_2 times, suggesting that this value is not significantly different from others in the literature. Therefore, either additional justification is needed to support this claim, or the statement should be rephrased to align more consistently with existing research.

Response 1: We agree that our T_2 time is comparable with the other spin-qubit candidates embedded in a nuclear-spin-rich host. We have toned down the discussion over the manuscript, without claiming the remarkable properties. We now more focus on searching for spin qubits in a material system of HDPs that are low cost, with the added benefits in the flexibility to control the spin and electronic properties of the spin centers, given a large structural and chemical library of the halide perovskites in general. We added this discussion on page 3-4 of the main text (marked in red).

2. The optical properties of the perovskite material are mentioned. Is this relevant to the study? If so, please provide justification.

Response 2: Optical properties associated with the atomic-like transitions of transition-metal (TM) ions are crucial for e.g. optical initialization and optical readout of spin qubits, as commonly employed in well-known solid-state spin qubits such as NV center in diamond and silicon vacancy in SiC. Therefore, we have now included a new section dedicated to the optical properties of Cr^{3+} photoluminescence in $\text{Cs}_2\text{NaInCl}_6$. Within this section, titled 'Optical Addressability of the Transition-metal Spin Qubits', we report new results identifying an atomic-like, spin-selective luminescence transition directly linked to our spin center. Based on them, a possible protocol is suggested for optical initialization and readout of the Cr^{3+} spin qubit.

3. What is the role of Indium? Would single crystals of $\text{Cs}_2\text{Ag}(\text{Fe/Cr})\text{Cl}_6$, $\text{Cs}_2\text{Na}(\text{Cr/Fe})\text{Cl}_6$ degrade more quickly if used instead?

Response 3: The In^{3+} ions, being diamagnetic (having no free electrons), serve as non-magnetic spacers for the transition-metal ions. This arrangement effectively reduces

detrimental spin-spin interactions between TM ions, which are known to harm spin coherence properties.

Indeed, increasing the concentration of Fe or Cr (e.g., by moving towards $\text{Cs}_2\text{Ag}(\text{Fe}/\text{Cr})\text{Cl}_6$ or $\text{Cs}_2\text{Na}(\text{Cr}/\text{Fe})\text{Cl}_6$ alloys) significantly shortens the T_2 coherence time. This effect is demonstrated in Figure R2.

Figure R2: the T_2 of the Cr[Na] samples with increasing Cr concentration by increased percentage of CrCl_3 molar ratio.

4. Figure 1A presents a schematic of the TM spin qubits embedded in spin-rich-host cubic halide double perovskites. However, the authors have not provided any supporting evidence for this for example from X-ray diffraction measurements or referenced any studies that include actual structural data for these crystals.

Response 4: We have now provided an appropriate citation to a previous X-ray diffraction study of the host $\text{Cs}_2\text{NaInCl}_6$ and $\text{Cs}_2\text{AgInCl}_6$, that confirms the cubic crystal structure (<https://www.nature.com/articles/s41586-018-0691-0>). The crystal structure file can be found at:

<https://next-gen.materialsproject.org/materials/mp-989571> ($\text{Cs}_2\text{NaInCl}_6$)

<https://next-gen.materialsproject.org/materials/mp-1096926> ($\text{Cs}_2\text{AgInCl}_6$)

5. The authors have mentioned the T_2 times for Cr and Fe in [Na] host. What are the corresponding numbers for Cr and Fe in [Ag] host?

Response 6: The T_2 for Cr[Ag] and Fe[Ag] is 5 and 4 μs , which is the same as compared with The T_2 for Cr[Na] and Fe[Ag] after calibrating the concentration of the TM ions to be on the similar order of magnitude shown in Extended figure 2 in the main manuscript. We also update the figure and provide the fitting in the new version of Extended figure 2B.

6. Supplementary material-supplemental, note 3; CPMG nuclear spin sensing experiment: The pulse lengths are not mentioned; Figure 5, “E “ is missing.

Response 6: We have corrected these issues in the revised paper.

Reviewer #3 (Remarks to the Author)

We thank Reviewer 3 for the valuable comments.

Reviewer #4 (Remarks to the Author)

The authors present a study of the electronic and nuclear spin coherence for transition metal centres in halide perovskite materials. The work presents coherent control of the electronic spins and claims entanglement of electron and nuclear spins. Spin coherence (T₂) of 29 us at 4K is reported, and the spin coherence timescale is attributed to the localisation of electronic density on the transition metal centre. The paper presents extensive characterisation, via various EPR techniques, of the hyperfine interaction between the central electronic spin and the Cr³⁺ and Fe³⁺ nuclei.

Response: We thank the reviewer for recognizing our extensive effort in coherence studies of electron nuclear spin system in halide double perovskites. Please find our detailed response to the reviewer's comments below.

My questions:

1. The work is motivated by the search for coupled electronic-nuclear spin systems for quantum information processing technologies. The authors compare the perovskite system with other 'disordered e-N' systems such as defects in diamond and SiC and self-assembled quantum dots (lines 44-45). However, these systems are very different to the one presented – they are singly optically-addressable electronic spins systems. Which means they have potential for applications that require spin-photon entanglement (ie. nodes for optical networking). It seems to me that the perovskite system is not comparable in this regard – it consists of a large ensemble of electronic and nuclear spins, and there is no mention of optical addressability. Thus, the authors should explain why it is relevant to compare to these other systems, whether it shows potential for application to similar technologies, and if not, what application they foresee for this system. The answers to this question may

Response: We apologize for not having clearly described earlier the transition-metal (TM) spin centers in halide double perovskites studied in our present work. These centers are in fact just like the defects in diamond and SiC. They are atomic impurity ions with well-defined intra-d-shell optical transitions, which are governed by spin-dependent selection rules such that they can be individually addressed. Such optical access provides the means for optical initialization and readout as well as the potential for spin-photon interfacing, similar to the defects in diamond and SiC. The main difference between the TM center in halide double perovskites and the defects in diamond and SiC is a stronger electron-nuclear spin interaction in the former with a nuclear-spin-rich host lattice, due to a stronger electron localization. This means a stronger perturbation of the electron spin by the nuclear spin environment, leading to spin decoherence. On the other hand, a stronger electron localization limits the number and randomness of nuclear spins (only within the nearest-neighbor ^{35,37}Cl and the next-nearest neighbor ¹³³Cs) that the electron spin can interact, making the corresponding electron-nuclear spin interactions well-defined and site-specific. This allows us either to avoid the non-deterministic dephasing from different isotopes like ³⁵Cl and ³⁷Cl, or to selectively control spin coherence using the non-dephasing ¹³³Cs nuclei. The latter can further drive electron-nuclear spin transfer, just like that demonstrated for the NV center in diamond between the electron spin and a nearby ¹³C nuclear spin.

The experimental evidence for the optical addressability of the studied Cr³⁺ spin center in Cs₂NaInCl₆ can be found in the new section of " Optical addressability of the TM spins in HDPs" on page 16.

We understand that, in terms of spin coherence properties, the TM center in halide double perovskites cannot compete with color centers in few-nuclear-spin or isotope-purified nuclear-spin-free diamond and SiC, nor is it meant to be. The competitive edge of spin centers in halide double perovskites rather center at their low-cost material synthesis as compared with the vastly more costly diamond and SiC, the availability of a large library of monovalent and trivalent cations (spin centers) without introducing excess charge noise, relative ease in incorporating spin centers in crystals without involving damaging processes with high-energy particles required to generate vacancies in diamond and SiC, superior and widely tunable optical properties covering a wide spectral range from UV to infrared, etc. Therefore, any potential spin components utilizing halide double perovskites are not meant for the highest performance, but they could rather find application areas where cost concern becomes paramount such as mass-production dispersible consumer electronics. At this early stage of spintronic research on this newly emerging materials system, the aim of our present work is to assess (1) whether the essential properties of spin centers in halide double perovskites are still within the acceptable range for quantum applications; (2) whether the nuclear-spin rich environment can be controlled and even harnessed for further applications. We hope that our study sheds light on design rules to identify promising candidates of spin centers and their perovskite host crystals, which could potentially provide a viable and competitive platform for low-cost quantum spintronics.

We have now included the above discussion in the revised Introduction section of the revised manuscript.

2. The authors claim to show entanglement of the electronic spin of the transition metal and the neighbouring nuclear spins. This discussion is focussed on the data shown in Fig 2(C). Here, the decoherence traces measured during the Hahn echo measurements is presented. The decoherence profiles show strong modulations due to the coupling to nuclear spins, as seen for other systems. The authors should explain why this constitutes demonstration of e-N entanglement. Entanglement should be confirmed via measurement of the density matrix of the e-N state (see Bourassa et al., Nature Materials, 2020).

Response We agree that our statement was inaccurate and that the observed modulations in the Hahn echo decay do not, in themselves, constitute a demonstration of entanglement. The reviewer is correct that such a claim requires more advanced measurements, such as quantum state tomography to reconstruct the density matrix. Unfortunately, our instrument limits such demonstration as we do not have a capability to perform electron-nuclear spin double resonance.

We have revised the manuscript to remove the claim of "entanglement". The purpose of our discussion on these decoherence traces is to highlight the presence of a strong, coherent coupling between the electron and nuclear spins. The long-lived modulation profiles in the Hahn echo decay provide concrete evidence that the electron spin is coherently coupled to a well-defined nuclear spin environment at the lattice sites. This coherent interface is a prerequisite for creating and manipulating entangled states.

3. With regards to the T_2 timescale, I agree that 29 μs is surprising for this system considering the nuclear spin composition. However, I am not sure if the factors that limit this coherence are well understood by the authors.

i. Why does T_2 show a relatively strong temperature dependence? A drop from 29 μs to 6 μs , from 4 to 20 K, is actually rather strong. I wouldn't expect decoherence that is dominated by nuclear spins to be strongly temperature dependent.

Response: We thank the reviewer for the valuable suggestion of the temperature dependence of T_2 . We have now performed a detailed temperature dependent study of T_2 and our findings indicate that the T_2 remains at a constant value below 10 K as expected from the hyperfine interaction that are temperature independent. The sharp rising of the decoherent rates above 10 K are dominated by the spin-phonon interaction, where the localized phonon frequency of about 50-70 cm^{-1} is associated with the activation of lower-lying phonon modes associated with Cs and Cl atomic motions.

Figure R3: Comparison between the temperature dependence of spin decoherence and spin relaxation rates in all four samples. Above 10 K the spin relaxation rate rapidly accelerated, dominating over the spin decoherence at $T > 20\text{K}$.

$$\frac{1}{T_1} = A_{dir}T + A_{loc,Cs} \frac{e^{\theta_{E,Cs}/T}}{(e^{\theta_{E,Cs}/T} - 1)^2} + A_{loc,Cl} \frac{e^{\theta_{E,Cl}/T}}{(e^{\theta_{E,Cl}/T} - 1)^2} \quad \begin{array}{l} \theta_{E,Cs} = 75 \text{ K} \\ \theta_{E,Cl} = 140 \text{ K} \end{array}$$

Figure R4: Temperature dependence and model fitting of spin relaxation rate based on localized spin phonon interaction.

ii. Are the modulation features in Fig. 2C consistent with a relatively weak hyperfine interaction (0.1-10 MHz)?

Response: Yes, this is consistent.

The modulation depth in an electron spin echo experiment is a direct result of electron-nuclear (e-N) spin mixing. This mixing allows normally forbidden EPR transitions—those where both the electron and a nuclear spin flip simultaneously—to become weakly allowed. The efficiency of this process determines how strongly the echo signal is modulated.

Analytically, the modulation depth, k , can be estimated by:

$$k = \frac{A_{nonsecular}^2}{\omega_I^2 + A_{secular}^2} \dots (1)$$

Here, ω_I is the nuclear Larmor frequency, and the secular ($A_{secular}$) and non-secular ($A_{nonsecular}$) terms are the orientation-dependent components of the hyperfine tensor in the lab frame.

The observed weak modulation, especially in the data from the powder sample shown in Figure 2C, can be attributed to two main factors.

First, the system is intrinsically far from the ideal conditions for strong modulation. The depth is maximized when the cancellation condition, $2\omega_I \approx |A_{iso}|$, is met. For our system's nuclei, the values are not close:

- ^{133}Cs : $2\omega_I = 3.9$ MHz, while $|A_{iso}| = 1.1$ MHz.
- $^{35,37}\text{Cl}$: $2\omega_I \approx 6.4$ MHz, $|A_{iso}| \approx 1.6\text{--}2.1$ MHz. Since this condition isn't met, deep modulation isn't expected even in a single crystal.

Second, the use of a powder sample further weakens the observed modulation due to orientation averaging. In a powder, all possible orientations of the crystallites relative to the external magnetic field exist simultaneously. The modulation depth k is highly dependent on this orientation angle, θ . For an axial hyperfine tensor, the components in Equation (1) vary with orientation as:

$$\begin{aligned} A_{secular} &= A_{\perp} \sin^2 \theta + A_{\parallel} \cos^2 \theta \\ A_{nonsecular} &= (A_{\parallel} - A_{\perp}) \sin \theta \cos \theta \end{aligned}$$

While some specific orientations might produce deep modulation, the total measured signal is an average over all angles. This averaging process washes out the sharp features, resulting in the weak, broad undulation seen in the final spin echo signal.

iii. Why are the modulations much stronger for the Cr complex than for the Fe complex?

Response: The hyperfine interaction involving the Cs atom is much greater for Cr^{3+} as compared with Fe^{3+} , which causes a stronger modulation.

iv. Could there be another factor that helps to decouple the electronic and nuclear spins in this system- possibly symmetry related?

Response: That's an excellent point. Yes, we agree with the reviewer that symmetry is indeed an important factor in both coupling and decoupling of electronic and nuclear spins in our system.

This high degree of symmetry is precisely what enables our experimental control over spin mixing in the single-crystal sample. By changing the orientation of the external magnetic field relative to the crystal axes, we can precisely tune the strength of the electron-nuclear coupling. This orientational control allows us to selectively turn the interaction with specific nuclear species "on" or "off."

This technique is the essential key to our method. It allows us to directly address and manipulate the ^{133}Cs nuclear spins while simultaneously decoupling the electron spin from the $^{35,37}\text{Cl}$ nuclear spins.

We have now included a discussion of this point in the sections on single-crystal CPMG and 3-pulse stimulated echo experiments in the revised manuscript.

4. Related to my question above, the authors state in the text that the strong localisation of the

electronic density on the transition metal is responsible for the long spin coherence timescale. This feels speculative and I don't see any evidence for this conclusion. Can the authors support this claim with modelling?

Response: We are grateful to the reviewer for this suggestion. Indeed, we can provide both experimental evidence and the modelling supporting our claim.

First, by experimentally comparing the electron spin decoherence between Na and Ag hosts, we show that the change in the third nearest neighbour (Na/Ag) does not affect the spin coherence time. This implies that the localisation of the electron wavefunction would be within the second-nearest neighbour nuclear spins. We have highlighted this discussion in the main text (in red) on page 7.

Second, we can estimate the degree of localization based on the Fermi hyperfine contact model, which shows that the electron wavefunction is indeed localized within the core of spin centres, effectively diminishing below 0.02 % at the ^{133}Cs sites.

We have included the corresponding discussion in the main text, page 12 and also in a separate section of 'Supporting Information.' The section reads as follows:

The Fermi contact interaction is a part of the hyperfine interaction of electrons in approximately spherical-like orbital and surrounding nuclear environment. It can be expressed by:

$$A_{iso} = \frac{2}{3} \mu_0 g_e \beta_e g_N \beta_N |\Psi|^2$$

This can also demonstrate the electron spin localization, where the percentage of electron density, $|\Psi|^2$, distributing at the neighbor atoms is estimated by calculating the ratio between electron density distribution caused by the hyperfine interaction and free electron localized at each nuclear site:

$$\frac{|\Psi_{hyperfine}|^2}{|\Psi_{free}|^2} = \frac{|A_{iso,experiment}|}{|A_{iso,free}|}$$

In our case, we consider 3 nuclear sites representing the nearest neighbours (^{35}Cl and ^{37}Cl) and the next-nearest neighbours (^{133}Cs) of the incorporated transition metal. By inserting the hyperfine coupling strength, we can obtain the percentage of electron localization at each nuclear site (See Figure S1 and Table S1).

Figure S1: The example illustration of electron density localization at each nuclear site ($^{35,37}\text{Cl}$, ^{133}Cs) around the incorporated Cr^{3+} .

Table S1:

Nuclear site	$A_{iso,experiment} = \frac{A_x + A_y + A_z}{3}$ (MHz)	$A_{iso,free}$ (MHz)	% electron localization
Cr[Na]			
¹³³ Cs	1.1	9.2x10 ³	0.012%
³⁵ Cl	-2.1	80	2.6%
³⁷ Cl	-1.6	66	2.5%
Na	-	-	-
Cr[Ag]			
¹³³ Cs	1.5	9.2x10 ³	0.017%
³⁵ Cl	-2.3	80	2.9%
³⁷ Cl	-1.8	66	2.8%
Ag	-	-	-
Fe[Na]			
¹³³ Cs	0.1	9.2x10 ³	0.001%
³⁵ Cl	-2.2	80	2.8%
³⁷ Cl	-1.7	66	2.6%
Na	-	-	-
Fe[Ag]			
¹³³ Cs	0.1	9.2x10 ³	0.001%
³⁵ Cl	-2.4	80	3.0%
³⁷ Cl	-1.8	66	2.8%
Ag	-	-	-

Additional comments:

1. The paper would benefit from a diagram of the electronic structure of the spin system.

Response We have added this in Figure 6A.

2. Line 99 should say degenerate, not degenerated.

Response We have corrected this accordingly.

Reviewer #4 (Remarks to the Author):

One of my main concerns about the manuscript in the first round was that it was comparing the perovskite system to single optically addressable spins in diamond, and other single spin-photon interface platforms. I stated that these are very different systems, because the former offer single qubits, whereas the perovskite system is an ensemble (of which there are many). I don't believe the authors understood or answered this question adequately. I can see reviewer 1 had the same concern. This means the motivation of the paper- and impact- is questionable. Despite the authors having retracted much of the language from the first round, I still feel the title, abstract and first paragraph of the introduction are misleading, and potentially confusing for people getting into the field.

For this reason, I can not support publication in Nature Communications in the current form.

Our response:

We thank the reviewer for his/her concern on comparing the perovskite system to single optically addressable spins in diamond and other single spin-photon interface platforms. We apologize for not having clarified this issue adequately in our previous revision.

A physical system of solid-state spin qubits is a solid material that contains an ensemble of isolated localized single spins where each spin can be individually addressed and entangled with another qubit, either another stationary spin qubit or a flying photon qubit through electrical or optical means. With a typical concentration of $< 10^{19}$, spin centers based on deep-level vacancy defects and TM impurities in diamond, SiC and perovskites remain electronically and magnetically isolated due to their strong localization of electron wavefunctions and spin densities. *Therefore, as a physical system, the spin-qubit system based on isolated TM ions in perovskites studied in our work is conceptually similar to the well-known color-center spin qubits in diamond and SiC whether they are vacancy defects or TM ions. Here, the notion of a single spin or an ensemble of spins does not refer to a fundamental limit in material classes but is merely a matter of how many spins are probed that is determined by the size of an experimental probe.* With a large probe size a large volume of a sample containing an ensemble of localized spins are detected, whereas a sufficiently small probe can focus down to a single localized spin even on the same sample. How small the experimental probe is required depends on the distance between adjacent spin centers and therefore its concentration. For example, with a confocal optical probe with a typical detection volume of $0.1-1 \mu\text{m}^3$, single-spin detection will require a concentration of the localized spin centers to be lower than 10^{12}cm^{-3} . Ideally, a higher concentration of spin centers is generally preferred in actual device applications that demand miniaturization and reduced material footprint. Here, on-chip integration with nanophotonic components for nanoscale light generation, transmission and detection need to be developed that is currently unavailable even for the technologically far-more advanced SiC.

We should also note that the incorporation of the TM ions in the perovskites remains within the doping range ($< 10^{19}$) in our study, far from the regime of alloying where TM ions can interact with each other. Therefore, these TM ions are predominantly isolated. For single-spin detection by a standard confocal optical microscopy, however, a lower doping concentration than that in the studied samples is required. Intense research efforts are under way to overcome technical challenges to significantly decrease doping concentration and to reduce the experimental probe size or/and sample volume (e.g. nanocrystals). We believe that single-spin detection is feasible in the near future, given recent advancements in fabrication technology for halide double perovskites.

The impact and motivation of our paper are to introduce this new class of chemically versatile and low-cost TM-doped HDP platform for spin qubits and to provide the first demonstrations that it possesses the essential quantum properties (long T_2 , deterministic nuclear coupling, and spin-selective optical transitions) required to be a viable candidate for a solid-state spin qubit system. Though limited to ensemble studies, we believe that the general knowledge of the spin physics gained in our study should be valid for single spin centers. For example, the spin coherence time measured from the ensembles sets the low bound for a single center. These ensemble-level demonstrations represent a necessary and critical first step to validate the platform before any future work to scale it down to the single-spin level is to be undertaken. *We should point out that the foundation of the currently highly successful single-spin studies in diamond and SiC is indeed based on earlier ensemble studies [e.g., Foehl, Buckley, Heremans, Calusine and Awschalom, Nature 479, 84 (2011)] that demonstrated the potential of these solid-state spin qubit systems and stimulated the intense follow-up research leading to many breakthroughs.*

We have now modified the title, abstract and introduction of our manuscript to clarify these aspects. We hope that the reviewer will now find our response and the revision satisfactory.